# Recent Advances in Skin Chemical Sensors

**DOI:** 10.3390/s19204376

**Published:** 2019-10-10

**Authors:** Benoît Piro, Giorgio Mattana, Vincent Noël

**Affiliations:** Université de Paris, ITODYS, CNRS, UMR 7086, 15 rue J-A de Baïf, F-75013 Paris, France; giorgio.mattana@univ-paris-diderot.fr (G.M.); vincent.noel@univ-paris-diderot.fr (V.N.)

**Keywords:** skin sensors, flexible, wearable, sweat, smart dressings, smart tattoos, diabetes, printed electronics

## Abstract

This review summarizes the latest developments in the field of skin chemical sensors, in particular wearable ones. Five major applications are covered in the present work: (i) sweat analysis, (ii) skin hydration, (iii) skin wounds, (iv) perspiration of volatile organic compounds, and (v) general skin conditions. For each application, the detection of the most relevant analytes is described in terms of transduction principles and sensor performances. Special attention is paid to the biological fluid collection and storage and devices are also analyzed in terms of reusability and lifetime. This review highlights the existing gaps between current performances and those needed to promote effective commercialization of sensors; future developments are also proposed.

## 1. Introduction

The new generation of healthcare sensors has developed quickly these last few years, both at research and commercial levels, essentially due to the emergence of personalized and preventive medicine. Indeed, there is a growing demand for analytical devices capable of performing a diagnosis at any time, preferably continuously. Such devices should ideally be wearable and, at the same time, also easy to use while being having connectivity to ensure prompt transmission and processing of the measurement results. The development of flexible electronics is rapidly increasing the ability to implement miniaturized components, allowing the engineering of ultra-thin and compact devices on conformable substrates, or even on transferable tattoos. In addition to connected wristbands able to measure heartbeat or even to perform physical marker monitoring, such as electrocardiogram (ECG), devices dedicated to chemical markers assay have developed as well. However, most of the current wearable chemical sensors are, for the vast majority, at the stage of proof of concept with no commercialized products. Currently, several breakthroughs are still missing to allow a true expansion of wearable chemical sensors commercialization; these breakthroughs are all mostly correlated to the need for contact between the device and the biological fluid to be analyzed.

Obviously, blood is the most interesting fluid to target as many of the analytes to assay are present in the largest possible concentration. Invasive sensors, in particular those featuring needles or even microneedles, have shown their analytical efficiency and have reached high level performances [1,2,3,4,5,6,7,8,9,10,11,12,13,14]. However, the future of diagnostics is said to rely on the non-invasive devices—i.e., devices that do not penetrate the skin—thus avoiding contamination and reducing infection risks and tissue injuries, while at the same time, improving the patients’ comfort. Therefore, current research has been focusing on the analysis of body fluids different from blood, such as tears, urine, skin secretions, and also breath [15,16]. 

Skin secretions can be broadly separated into two categories depending on their mode of transport through the derma: sweat and perspiration gases. Along with water (99% w/w), several metabolites are evacuated by sweating: ions (Na^+^, K^+^, NH_4_^+^, Cl^−^), antibodies, urea, uric acid, lactic acid, or other compounds directly related to our diet: glucose, alcohol, medicines, or even drugs. Besides these chemical elements, there is a non-zero leakage of chemical compounds from blood vessels to the dermal layer, a phenomenon called ‘perspiration’. Acetone, CO_2_, dimethyl methylphosphonate, ethene, but also water vapor and oxygen are compounds normally present in blood and are emitted through the skin. 

Besides the numerous chemicals that can be found at its surface, skin is also particularly interesting for its physical properties. Indeed, skin is an organ by itself, more precisely the largest organ of the human body; for an average man, it has a surface of approximately 2 m^2^. Hence, skin can accomodate placing relatively large devices onto the body without causing significant discomfort to the patient. 

In summary, due to its large surface area and chemicals present in its secretions, skin is undoubtedly an interface of interest in evaluating the presence of chemical markers in the blood. One should note, however, that the amount of sweat available for analysis can be significantly lower than tear fluid in the eye, and orders of magnitude smaller than the saliva volume available in the patient’s mouth. In addition, the following major drawbacks limit the development of analytical systems dedicated to skin analysis: (i) analyte concentration in sweat is not identical to the same analyte concentration in blood and rigorous correlations have to be established; (ii) the skin state has a temporal variability which depends on diet, temperature, hydration, age, gender, or even skin pathologies so information on analytes presence and quantity based on skin analysis is not always reliable; and (iii) skin is highly elastic with a Young’s modulus ranging from 10 kPa to more than 100 kPa, thus requiring the development of highly conformable devices to ensure the patient’s comfort, as well as a large contact area between the sensing part and the skin. All these factors obviously slow down the development of reliable skin sensors able to monitor an analyte present in blood accurately and continuously. Associated with the problem of correlation between markers content in blood and onto the skin, the analytical accuracy of the device is also closely related to the sampling methodology (e.g., accumulation of sweat, transdermal measurement, etc.) as well as the device reusability. Indeed, continuous measurements require that the sensor have no hysteresis and continue to operate in spite of the possible accumulation of analytes as a function of time. In that sense, sweat analysis concentrates all these problems as sweat is abundant during sport but clearly insufficient in normal activity requiring, in such a case, accumulation and storage and a consequent delay before measurement, as well as the necessity of developing a fluid controller apparatus. Moreover, sweating leads to an important accumulation of the aforementioned chemical markers at the contact interface between skin and device that can distort the subsequent measurements. Due to this last issue, disposable devices are certainly an efficient outcome, but they first require the use of biodegradable or at least eco-friendly materials and, second, a low-cost production. This last problem is currently being overcome thanks to advances in fabrication processes, in particular screen-printing and inkjet-printing techniques which allow for multicomponent integration on conformable substrates [17].

Several very well-documented reviews are currently available on the topic of wearable skin chemical sensors. Liu et al. in 2017 [8] defined the “lab on skin” concept, focusing on both the strategies of skin integration (temporary tattoos, hard/soft integration, and functional substrates) and device powering sources. Wu et al. reviewed in 2018 [18] the (nano)materials currently used in wearable sensors and the same year, Bai et al. [19] reviewed all patchable devices interfaced with the skin, with a focus on materials and fabrication techniques, including electrical, thermal, and mechanical sensors, which are not reviewed herein. Heikenfeld et al. [20] also reviewed such sensors, including optical ones, emphasizing the fundamental physiological mechanisms occurring in skin. In 2019, Xu et al. [21] enriched the discussion with a review on wireless systems. The domain being in constant and rapid development, it was necessary to focus once more on it, but from a complementary point of view; the latter review was indeed focused essentially on correlations between blood concentrations and sweat concentration, and on the sensors’ ergonomics.

To date, the development of transduction strategies makes it possible to have effective analytical tools, but the issues related, on the one hand, to the sampling method and, on the other hand, to the biodegradability of the used materials are rarely addressed. Keeping in mind these two aspects, we have reviewed here very recent (within the last three years) advances in skin chemical (bio)sensors, i.e., ideally thin, flexible, and wearable chemical sensors designed to detect and analyze substances related to skin state (e.g., water content, porosity, microbiome), aggression by external factors (e.g., chemical or biological pathogens), and skin secretions whose content may be correlated to blood content (e.g., ions, metabolites, etc.). Few non-wearable skin sensors are also reviewed, for those specific applications where wearable sensors are not available yet. 

We decided to organize the review by practical applications, so that the reader can directly refer to the application of their choice. Each analyte is different and technical solutions cannot be transferred directly from one analyte to the others, so that it was judicious to analyze the state of the art by analyte and not globally. Five chapters will be then dedicated to: (i) sweat analysis (for glucose, lactate, pH or other metabolites), (ii) skin hydration, (iii) skin wounds, (iv) perspiration of volatile organic compounds (e.g., alcohol or drugs), and (v) general skin status including microbiota. For each application, the most pertinent transduction methods will be presented, and originalities of each approach underlined. Furthermore, attention will be paid to the sampling method as well as the ability of the device to perform multiple measurements without loss of accuracy.

## 2. Glossary of Acronyms

To keep the review easy to read, many acronyms are used. For the sake of clarity, a glossary of acronyms is provided below (Table 1), sorted by alphabetical order.

## 3. Discussion

### 3.1. Sweat Analysis (Most Common Applications)

#### 3.1.1. Multicomponent Analysis of Sweat

Sweat is a colorless liquid secreted by the sudoriferous glands found in the skin and emitted on the epithelial surface by way of ducts. In humans, sweat evaporation (called ‘perspiration’) is the predominant heat dissipation mechanism [22]. Sweat contains mostly water (up to 99%) but also a small amount of solutes such as, by (decreasing) concentration order: sodium, potassium, calcium, magnesium and chloride ions, lactic acid and urea, iron, zinc, copper; and sometimes also chromium, nickel, and lead ions; as well as some endogenic (e.g., glucose) and exogenic organic compounds. Since many analytes in sweat are electroactive molecules and/or ions, sweat analysis are often based on electrochemical transduction. Several reviews were published, for example by Gao et al. [23] who reviewed in 2016 wearable perspiration biosensors for biomarkers present in sweat, by McCaul et al. [24] in 2017 who emphasized on the platforms dedicated to the monitoring of electrolytes in sweat, or by Bandodkar et al. [25] or Bariya et al. [26] in 2018. One must consider that sweat is the main matrix for skin sensors, far before the others (wound exudate or volatile organic compounds in perspiration).

The literature shows two main approaches for the transduction principle: electrical or colorimetric. For the first approach, it is necessary to properly design the electrical tracks and connections so that they can resist stretching, bending, or any other deformation to which the device could be subject. This difficulty overcome, signal treatment and data transmission are easy to perform, which make this solution particularly potent. The second approach is easier because the result is visualized by eyes or indirectly using a camera and no electronics need to be embedded within the wearable device. The drawback is that colorimetric assays are generally poorly quantitative. As an example of ab electrochemical device, Ashley et al. [27] focused their work on the geometry of the conducting tracks onto a highly conformable electrochemical patch made of polyimide (PI) and silicone (Figure 1), the final goal being the creation of sensors capable of adapting to all curvatures of the human body and resisting to the natural mechanical stresses applied to the skin while, at the same time, allowing the mass transfer of gas and fluids from the skin surface to the sensing material. The basic idea is that the conductive tracks are highly curved (coiled) to be stretchable in every direction. Taking good electric contacts, as for any other flexible electronic devices, is indeed a challenge by itself. No measurement was made directly on the skin, however, and the electronic setup used for data collection was relatively heavy and not integrated into the patch.

To avoid the difficulties generated by embedded electronics, Choi et al. [28] developed colorimetric assays integrated into a multi-instrumented PMMA patch for measurements of sweat rate, pH, temperature, concentrations of ions such as chloride and other metabolites such as glucose and lactate (Figure 2). This patch obviously does not require any power supply or electric wiring; the downside is that the results are mostly qualitative and data transmission is possible but only using external systems/devices taking pictures such as smartphones (Figure 2E). Except for the chlorine assay which uses a long channel for fresh sweat collection, other assays use collection microreservoirs of 5 μL. It is also important to note that the sensor is not reusable in its actual state.

This strategy which consists in using colorimetric indicators integrated into sensing areas of a wearable patch, then taking a picture of the patch with a smartphone including picture treatment software able to translate the different color grades into quantitative values of analyte concentration has also been proposed by He et al. [29] (for a glucose sensor). Here, fresh sweat is collected with the help of a large (the size of the patch) superhydrophobic surface which guide the liquid to the test area. As for the previous example, it is not reusable. These approaches are not fully satisfactory, due to their lack of selectivity and the large errors in estimated analyte concentration. Taking these drawbacks into consideration, Bandodkar et al. reported a wearable sweat sensor which relied either on electrochemical or colorimetric transductions [30] (Figure 3), therefore benefiting from the advantages of the two transductions. Interestingly, while power source is a major preoccupation in wearable electronics, no power supply was necessary in this example because the device was powered by contactless near field communication (NFC). The authors demonstrate in-situ, semi-quantitative, monitoring of sweat rate and pH; as well as concentration of lactate, glucose, and chloride. One should note that NFC powering has a drawback, however, because the sensor is active only when the smartphone is put in close contact with it; for the rest of the time, no data can be acquired. As a consequence, it cannot be considered as a continuous monitoring system. Also, as reported by the authors, electronics are reusable but not the collection/sensing part, which must be changed for each set of measurements.

Most of the devices we reviewed are the so-called ‘wearable devices’, i.e., devices and systems that can be directly incorporated into clothes or put into direct contact with the skin as accessories, body implants, or even garments.

Gualandi et al. [31] published in 2016 the development of wearable chemical sensors made on textile for the detection of redox active molecules (adrenaline, dopamine, and ascorbic acid) in sweat. The sensors are based on very simply designed organic electrochemical transistors (OECTs) (Figure 4A) entirely made of conductive polymer (PEDOT:PSS) and deposited on the fabric (woven cotton and Lycra) by screen-printing (Figure 4B). Devices fabricated on these substrates present performances comparable to those of OECTs fabricated onto more classical non-conformable plastic substrates. Addition of small droplets of an electrolyte solution (mimicking human sweat) on one side of the fabric is enough to produce reliable and reproducible measurements, as if the devices were in direct contact with the skin perspiration on a garment. The operating voltage is low (<1 V) as well as the power needed (~10^−4^ W). In this study, no experiments were performed with interfering species or with real sweat; however, the devices are not equipped with a sweat collection system and their reusability was not evoked. This work was fundamental in the field because it showed that efficient organic transistors can be printed on textiles and still keep excellent electronic properties. Textile being the material of our garments, it means that electronic functions can be integrated directly on them during the industrial fabrication process. It also demonstrates the potential of OECTs for skin sensors, where the so-called ‘ions-to-electrons’ mechanism, typical of OECTs, can be fully exploited. The advantages of transistors will be discussed later in this review.

Textile is not the only seducing substrate, however. Tattoos, obtained by transferring onto the skin of a shape previously printed on a thin foil, are particularly interesting. Indeed, such tattoos allow a high deformation ratio when made of very thin and stretchable materials such as silicones or made in a shape that intrinsically bears a high deformation ratio. This is the case for Gao et al. [32] who reported an original highly stretchable tattoo inspired by fish skin (Figure 5) into which they included photonic crystals for fluorescence on-skin sensing of lactic acid and urea contained in sweat. The authors, by choosing optical transduction, overcome the difficulties of integrating electrical connections on such highly structured substrate but, conversely, they obtain devices that are only poorly quantitative and not reusable.

On-body collection of sweat is a challenge by itself, for at least two reasons. The first is that sweat is only produced in perceptible quantities when the subject is doing sport or at least a sufficiently long physical exercise, and when the temperature is sufficiently high; in other situations, no sweat can be collected at all. The second reason is that collecting sweat means to drive sweat from the top of the skin to the sensing areas in a controlled manner, through channels, without mixing or losses of liquids. To overcome these difficulties, Anastasova et al. [33] published in 2017 the development of a thin flexible wearable patch that incorporated paper-based microfluidics. The patch consists of several layers, as shown on Figure 6A; to ensure a continuous capillary flow of sweat adjusted by tuning the papers absorption rates: Whatman 113 at the sweat collectors (pads) and Whatman 4 in the straight channels going to the sensors. No more than 15 μL of sweat are needed to wet the device from the pads to the sensors. The system can measure lactate, pH, and sodium cations (Figure 6B–D); it also contains temperature sensors used for internal calibration. Sensors are conventional, with sodium ions quantified potentiometrically using a PVC membrane deposited on a layer of PEDOT, pH sensing based on an iridium oxide/Pt electrode and lactate with lactate oxidase enzyme entrapped in a polyurethane (PU) membrane. The most interesting part of this work is the design of the patch. The sensitivities of the sensors are comparable to the state-of-the-art, with signals reaching steady state within 10 s; even if not fully reusable, the devices allow real continuous monitoring, with sweat inlets and outlets, with a satisfactory time resolution, which makes them one of the best performing sweat analysis platforms described to date.

Not using paper, but surface tension-driven microfluidics silicone-based stickers, in 2017 Martin et al. [34] described a non-invasive sweat monitoring flexible patch obtained by lithographic and screen-printed technologies, for electrochemical monitoring of glucose and lactate. The device enables natural pumping of fresh sweat to the electrode chamber, at a rate slightly lower than 1 μL.min^−1^ (Figure 7), which was sufficient for the analysis. It must be kept in mind that such flow is only possible if the subject is placed under conditions where their skin actually perspires; silicone patches, which are totally impermeable, make this sweating possible even without significant physical effort from the wearer. As for the previous example, the devices were tested in real conditions and were able to collect then eliminate sweat, so that continuous monitoring was possible. Reuse was not discussed by the authors.

These previous examples show the various possibilities in terms of transduction, sweat collection, shapes, and materials. The following section concerns applications to diabetes monitoring, which is by far the biggest market for such sensors.

#### 3.1.2. Diabetes Monitoring

Diabetes affects hundreds of millions of people all over the world. The World Health Organization mentions a real epidemic with an estimated number of cases which has grown from 30 million in 1985 to an expected number of 300 million by 2025—i.e., more than 6% of all adults over the world. For those who suffer from this disease, frequent monitoring of blood glucose level is mandatory. Blood glucometers have been on the market for more than 30 years and today, wearable continuous glucose monitoring devices (CGM) are also available (Dexcom G5^®^, Inc. San Diego, CA; FreeStyle Libre, Abbott Diabetes Care, Alameda, CA; Medtronic Enlite^®^, Minneapolis, MN). These CGMs rely on subcutaneous needles in contact with the interstitial fluid, implanted for 1 or 2 weeks maximum, and carrying amperometric enzymatic electrodes. They are reliable, but the presence of needles makes these systems invasive. This is the main reason why non-invasive sweat glucose monitoring techniques were developed. Kim et al. recently published a review [35] on such needle-free devices, showing the advantages of reverse iontophoresis over skin interstitial fluid and direct analysis of sweat over classical CGM equipped with a needle. It is acknowledged that glucose levels in interstitial fluid (ISF) follows very reliably blood glucose level, with only a few minutes delay. To have access to skin ISF without needles or microneedles, reverse iontophoresis was used. The GlucoWatch™, a glucometer briefly commercialized, exploited this technology (Figure 8A–C). While this device was robust, capable of continuous monitoring, and performed very well in real conditions, skin injuries appeared after repetitive use, due to the iontophoresis process repeated many times on the same spot, and the device was withdrawn from the market.

Optimized devices are still developed, however, as illustrated by the work of Emaminejad et al. [38]; see Figure 8D. One cannot forget citing one of the most convincing achievements in terms of electrochemical patch and iontophoretic extraction of the interstitial fluid. Indeed, in 2018, Kim et al. [39] published an appealing ‘Panda-looking’ device based on sweat stimulation via iontophoretic pilocarpine delivery and, symmetrically, iontophoretic extraction of ISF, including flexible electronics (Figure 9A–C). The stimulator/sensor is designed to be tattooed (Figure 9D). Performances were validated in real conditions, collecting fresh sweat from subjects consuming food and alcoholic drinks, with a good correlation with the corresponding blood levels (Figure 9E). In addition to glucose, the authors measured ethanol as well. Electrodes were conventional amperometric ones and the transduction principle was similar for both. The device, which did not need sweat to be extracted and stored for analysis, was capable of continuous monitoring. The effect of the iontophoretic extraction was not evaluated for long-term use, however.

Access to sweat, as seen previously, is more straightforward however, and it was shown more recently that sweat glucose levels can be also exploited to satisfactorily monitor blood glucose levels, which motivated a great number of developments on wearable, flexible patches able to collect sweat, measure glucose and transmit corresponding data (Figure 10).

In 2017, Cho et al. [42] reported a wearable paper-based glucose sensor able to analyze sweat, based on a glucose/oxygen enzymatic fuel cell integrated into a Band-Aid patch. This architecture allowed glucose monitoring in sweat without any need for an external power source, as for qualitative colorimetric patches, which obviates the traditional drawback of wearable electrochemical patches. As for other paper-based patches, proper elimination of used sweat was not discussed as well as the long-term stability of the device. Following another strategy, Han et al. [43] developed a multifunctional flexible electronic skin sensor on what they called a “piezo-biosensing unit” made of enzymes deposited on ZnO piezo-nanoarrays able to provide the necessary power for a potentiometric detection (the authors, at this stage of development, did not focus on strategies to collect and eliminate sweat, and did not discuss reusability). The device can detect glucose in perspiration, as well as other important biomolecules such as lactate, uric acid, and urea. Even if the piezo-enzyme coupling concept has to be better understood, this example is again a promising approach to overcome the difficulty of integrating batteries in thin flexible devices (even if printed thin flexible batteries already exist). Also in 2017, Abellan-Llobregat et al. [44] published a work on a flexible and stretchable patch for amperometric glucose monitoring in sweat, obtained by layer-by-layer screen-printing of several functional inks (for conductive tracks, working, counter, and reference electrodes) on a PU membrane. The device was not tested on real skin, however, but on real sweat samples collected before-hand; the authors focused on the sensors performances in particular, but not on the way to collect/eliminate sweat and therefore not on the ability of the sensors to perform continuously. A clear correlation between glucose concentration in perspiration and glucose concentration in blood (obtained from a commercial glucometer) was definitely demonstrated, which validated the hypothesis that sweat can be used as a reliable matrix for indirect blood glucose monitoring. Karpova et al. obtained the same results in 2019, with another sensing platform (as for the previous example, not optimized in terms of sweat collection, and which did not work for continuous monitoring) [45]: they found a Pearson correlation coefficient r = 0.75 (Figure 11A) between glucose concentrations in blood and in sweat and, better, an excellent correlation between the kinetics of these concentrations (Figure 11B). To obtain these results, they realized their monitoring on about 20 patients which showed consistent results with vein blood glucose monitoring.

Another aspect which has to be investigated for developing efficient wearable sensors is the materials: substrate material but also active materials. Zhao et al., for example, focused on this aspect [46]. In this work, the authors did not include their electrodes in a sweat collection patch and did not investigate real conditions but focused on the sensing materials. They developed a fiber-shaped enzymatic electrode of particular interest for integration in smart textiles. Their electrode is based on gold fibers embedded into a soft copolymer matrix of styrene–ethylene/butylene–styrene (SEBS) and functionalized with glucose oxidase. This textile biosensor achieved excellent sensing capabilities, insensitive to strain up to 200%. A similar approach was followed by Oh et al. [47] who also focused on encapsulation of the active material (CoWO_4_/CNT) into sticky silicone (Silbione™), which allowed them to investigate their sensor on fresh sweat collected directly on skin; continuous monitoring was performed, but electrodes sat directly onto the skin, without chambers or sweat collection channels. The active material demonstrates excellent mechanical stability while keeping its conducting and catalytic properties up to 30% stretching and showed good adhesion to skin over time. Martin et al. [34] described a flexible epidermal microfluidic detection device fabricated using both lithography and screen-printing, for sweat sampling and real-time enzymatic electrochemical monitoring of glucose and lactate levels. The flexible and wearable device integrates fluidics, electrochemical sensors, and flexible electronics for wireless data transmission. After only 8 min from starting the exercise, the device is able to send its first measure. In addition, it is resilient upon continuous mechanical deformation (Figure 12). This is another example of a very efficient device, working perfectly well in real conditions.

#### 3.1.3. Lactate Monitoring

Lactate monitoring is essential to follow the correct oxygenation of sportsmen during an intense effort. It is also of importance for critically ill patients, for whom overproduction of lactic acid (hyperlactatemia) may lead to death. Before discussing wearable device, one can cite the work of Nagamine et al. [48] who described in 2019 a hydrogel-based touchpad for lactate determination in sweat. The sensor by itself is a conventional electrochemical enzymatic (LOx) lactate biosensor, covered with an agarose gel in a phosphate buffer saline solution. When human skin—e.g., a fingertip—contacts the agarose gel (0.5 mm thick to decrease the response time), lactate contained in sweat diffuses into the gel, then leads to a change in potential at the underlying lactate oxidase-modified electrode. The device is therefore not designed for continuous sweat monitoring and does not feature any microfluidics. (Figure 13).

There are many publications concerning wearable devices for lactate monitoring, often based on the same principles used for glucose monitoring because the enzyme lactate oxidase, able to selectively oxidize lactate, is very similar in its functioning to glucose oxidase. As an example, one can consider the work of Jia et al. in 2013 [49] which is extremely close to the other approaches that was followed later by the same group for glucose monitoring. This device is a simple printed 2D sheet without fluidics; sweat uptake is performed simply by placing the sensor in contact with the skin (see previous section). Luke et al., more recently (2018) [50], described a wearable sensor consisting of a flexible Kapton^®^ foil coated with an adhesive and carrying an organic electrochemical transistor (OECT) functionalized with LOx. The interest of using a transistor is to considerably amplify, at the drain electrode, the current generated by the electrochemical processes occurring at the gate electrode of the OECT (which is the very same reaction happening on the working electrode of a conventional amperometric sensor). However, the sensing range of their device was limited to lactate concentrations below 1 mM, i.e., far below values found in real sweat after exercise. This illustrates the fact that, for lactate sensors as well as for glucose sensor, which are present in body fluids in relatively high concentrations, the limit of detection (LOD) is generally not the limiting factor, the operational range of concentration being much more important (LOD has a far higher importance in environmental and industrial sensing.). The sensors described in this last publication were built on a conformable substrate along with the electronics for signal treatment but did not feature any fluidic systems and direct measurements on skin were not reported.

On the contrary, the device developed by Hong et al. [51], which did not include a complex sweat collection system, was evaluated in real conditions. In this case, devices are composed of a strip integrating both a lactate and a glucose sensor, in order to combine the two results: not only are they able to measure glucose levels but they can determine if the subject is doing an intense effort (which results in lactate production) or not. In conjunction with other physiological data (skin temperature, heart beat), this approach helps to better predict the evolution of blood glucose not only as a function of time but also as a function of the physical situation of the subject, i.e., pre- or post-exercise, which allows much more efficient prevention of hypoglycemic shocks. With the same idea of monitoring lactate levels and their variations, Enomoto et al. [52] were able, using a sweat collection device (in the form of a simple impermeable plastic foil with a sweat outlet attached on the right upper arm of subjects) to quantify the dynamics of lactate secretion, which they found of ca. 4.50 μg cm^−2^ min^−1^ under high intensity exercise (heart beat: 160 bpm). The main measuring device is included in a relatively heavy remote fluidic system which is not designed to be portable. Tojyo et al. [53], on their side, reported a fully wearable device based on the same working principle as Enomoto et al., but with a portable microfluidic part put in direct contact with the skin. They also reported dynamics of lactate secretion, but of ca. 60 μg cm^−2^ min^−1^. These two results seem to demonstrate that secretion rates of lactate in sweat are highly subject-dependent and should be considered with care.

To conclude this section, it is interesting to discuss the work of Lei et al. [54], who reported on an original way to make highly flexible conductive tracks on wearable patches. Instead of screen-printing or inkjet-printing a metal ink, or patterning gold electrodes by conventional photolithography, they engraved channels into a silicone rubber substrate, which they later filled with a ‘liquid metal’ (actually, a liquid-metal alloy, provided by Changsha Santech Materials Co., Ltd., China) capable of good electrical conductivity while being liquid and therefore indefinitely bendable, stretchable and rollable (Figure 14A,B). The design of the sweat collection layer (Figure 14C) was also original, forming a kind of crosslinked network of coiled channels instead of the straight channels commonly reported in the literature; this was reported to improve sweat collection. In terms of sensing electrodes, these authors incorporated an original MXene/Prussian blue (Ti_3_C_2_Tx/PB) composite, designed to be flexible, conductive and efficient for electron transfer from the enzymes. Sweat collection was performed through an ad-hoc absorption layer in direct contact with the skin, made of complex patterns for optimized uptake efficiency. This device was evaluated for continuous monitoring, in situ, but for no longer than 30 min (Figure 14D).

#### 3.1.4. Ions Monitoring (Including pH)

Determination of some anions and cations in sweat has several health applications. Indeed, it allows detection and monitoring of diseases where too few or too many sodium or potassium cations are present in blood (and by consequence, in sweat), with important muscular consequences. Some heart failures can be detected this way, as well as cystic fibrosis. Determination of electrolytes in sweat has also major applications in sport or in everyday life. For example, the measurement of sodium and chloride ions in sweat is a good indicator of the dehydration status of a subject (simply, an increase in Cl^−^ or Na^+^ ion concentration in sweat indicates that there is less water available to evacuate these ions). Continuous monitoring of these ions may help provide personalized hydration strategies for sportsmen, but also for elder people who often suffer from dehydration without knowing it. Wearable sensors for ion monitoring have been extensively reported. The readers who want to further investigate this application field may refer to the recent and very complete review by Parrilla et al., published in mid-2019 [55]. In this section, we will only give significant examples which illustrate particular strategies.

Most wearable ion monitoring platforms rely on potentiometric transduction. However, as discussed previously, electrochemical technologies imply the embedding of electronics within the wearable device, which obviously increases the technical difficulty and even much more the production costs. Therefore, colorimetric devices, even if subject to drawbacks, keep their interest. In a similar way to works published by Choi et al. [13,33], Kim et al. [56] published the development of a colorimetric patch for determination of chloride ions in sweat. In this publication, they did not discuss the details of the colorimetric sensors, based on well-known reactions. On the contrary, they mostly focused on the microfluidic part, only driven by capillary forces. The originality here is that they triggered fluid flow within the various channels of their patch using valves made of a superabsorbent polymer (SAP) whose role is to open or close openings as a function of its swollen or collapsed state (Figure 15). This device was not designed for continuous monitoring and was not reported to be reusable.

However, electrochemical sensing protocols remain more quantitative than colorimetric ones. In 2015, Kim et al. [57] reported a tattoo-like electrochemical sensor for non-invasive monitoring of Zn^2+^ in sweat. Sweat collection was not particularly investigated, with a simple 2D impermeable sheet carrying planar electrodes and put in direct contact with the skin. The detection principle was based on the well-known stripping technique (usually used for detection in polluted industrial waters, however, and not for such applications) using a bismuth/Nafion electrode (and therefore not a potentiometric technique). The temporary tattoo sensor withstood repeated mechanical stress and displayed a well-defined Zn^2+^ response during on-body testing. A zinc concentration of 0.34 μg mL^−1^ was detected from the sweat of a subject. Continuous monitoring was not reported.

On the same technical basis of Martin et al. [34], Sempionatto et al. [58] described a potentiometric fully flexible patch for electrochemical monitoring of Na^+^ and K^+^ in sweat. The efficiency for monitoring sodium and potassium ions with this device was relatively deceiving, however, (Figure 16) and the most interesting approach of this work was the way sweat was efficiently collected, with inlets, channels, and an outlet (see Martin et al. [34], for details).

In 2016, Gao et al. [59] reported a very convincing mechanically flexible and fully integrated sensor array for on-skin measurement. No microfluidics or particular strategies for sweat uptake were developed in this work; the real breakthrough of this work was that all the components (sodium and potassium electrodes but also glucose, lactate, and a temperature sensor for calibration of the other sensors) were integrated together into the same substrate. In that sense, their device bridged the technological gap between already well-known biosensors, signal conditioners, and wireless transmitters, by putting them together on a single device made of a flexible polymer substrate and included a real flexible circuit board onto which conventional silicon electronic components were picked-and-placed (Figure 17A–C). Sweat profile of subjects doing physical activities were acquired; they were able to clearly monitor the hydration state during effort (Figure 17D).

The same year, Parrilla et al. published a textile-based, stretchable, potentiometric sensor array [60] for simultaneous K^+^ and Na^+^ monitoring, on the basis of a relatively simple fabrication process (electrodes printed on a textile substrate) and relying on standard ion-selective potentiometric electrodes (except for the fact that polyurethane was used instead of common PVC for the ion-sensitive membranes—Figure 18A). Only the sensor part was flexible, taking profit of stretch-enduring screen-printing inks along with a serpentine design of the conductive tracks. The device was accompanied by a relatively large power and data processing unit, however, wired a few centimeters apart from the sensor, which made the whole device relatively cumbersome (Figure 18B), and only ex-situ experiments were performed for the qualification of the sensors. The same group (Parilla et al., [61]) has recently reported a different version of the same system for pH, Cl^−^, K^+^, and Na^+^ embedded in a single flexible sampling cell instead of a simple textile, including channels for sweat collection and renewal. The electrodes were fabricated with stretchable materials which kept their properties long enough for midterm exercise practice (drift did not exceed 0.5 mV h^−1^), exemplified in this article by continuous monitoring over a period of 1 h.

Still featuring conventional potentiometric sensors, Hoekstra et al. [62] reported a wearable sensor for monitoring cations in sweat. The sensor was particularly simple. Transduction was performed by measuring changes in Donnan potential across a cation-selective Nafion© membrane simply cast onto a paper substrate. Electrical contacts were fabricated with a screen-printed carbon-ink. They used their device to acquire in real-time the conductivity profile of an athlete’s sweat during exercise, by putting the electrodes in direct contact with the skin, without any use of fluidics. Data transmission and visualization was performed on a smartphone application connected via Bluetooth^®^. Also focused on data transmission, Dang et al. [63] developed a wireless, flexible, and wearable system for sweat pH monitoring with good resistance to bending and stretching. The authors emphasized on the fabrication protocol leading to such flexible circuits (Figure 19), particularly the RFID stretchable antenna. They used a flexible PI film covered by a Cu layer which they blade-cut to obtain the antenna pattern and transferred it, with a water-soluble glue, to a planar substrate (a silicon wafer) onto which PDMS was spin-coated. After curing, the PDMS layer was detached from the silicon wafer to obtain a self-standing stretchable antenna. This antenna and the pH sensor (graphite as sensitive layer) were able to withstand more than 500 cycles to 30% strain before breaking. Its radiating performance was stable under 20% strain. The device was not tested directly with real sweat nor put in direct contact with skin, but it was employed to transmit the pH value of a human sweat equivalent solution to a custom-developed smartphone application.

This section cannot be concluded without citing original works which focused more particularly on the electrodes themselves. As shown above, most devices use very conventional potentiometric electrodes. This technique is very robust indeed but suffers from two drawbacks: the need for a reference electrode, which is never stable and induces drift; and relatively poor sensitivity due to the intrinsic logarithmic dependence of the potential on ions concentrations. A promising sensing strategy to overcome these difficulties is the use of ion-sensitive field-effect transistors (ISFETs), which have existed for 40 years but have not been reported so often in the field of wearable technologies. As an example, however, Nakata et al. [64] reported in 2017 a wearable, flexible sensor sheet for pH measurement on skin, using an ISFET, integrated with a temperature sensor (Figure 20). Most importantly, the authors demonstrated that their ISFET kept 100% of its drain current even after more than 1500 bending cycles. Their sensor was evaluated in-situ, on skin, with fresh sweat, but no particular care was taken for sweat collection and elimination. More recently, devices such as IS-FDSOI (ion-sensitive fully depleted silicon on insulator) were reported. Indeed, integration of such thin-film transistors on flexible foils is straightforward, and power need is minimal. Following this strategy, Bellando et al. [65] focused on the integration of IS-FDSOI with nanofluidics in SU-8 resin. They called their device the Lab on Skin™, able to simultaneously measure pH, Na^+^, and K^+^ in sweat. Each sensor is ultra-low power, consuming less than 50 nW (the particular advantage of SOI devices). Considering the high level of miniaturization, the authors designed the sweat inlets so that their size and special distribution fitted with human skin pores size and distribution, thus requiring zero energy for fluid pumping, which was exclusively based on capillary forces. They obtained flow rates of tens of pL.s^−1^ and actually tested their devices on skin to validate their efficiency. To end, a last remark concerns the stability of sensors using reference or pseudo-reference electrodes. It is indeed well-known that Ag/AgCl pseudo-references are easy to fabricate but degrade with time. Strategies to regenerate the reference are known and have already been applied, but these aspects were not discussed in the reviewed work.

### 3.2. Skin Hydration

The measurement of skin hydration is very important because hydration influences many essential functions of the skin such as mechanical resistance and barrier function. As a consequence, skin hydration plays also a role in the efficiency of drug delivery through the skin, and therefore in the efficiency of chemical skin sensors. In addition, it also plays a role in visual appearance and is therefore an important parameter considered by the cosmetic industry. Indeed, skin being our external envelope, it is a material of choice for the cosmetic industry, which is developing companion diagnostic to monitor the skin of its customers and thus propose high added value personalized solutions. Skin hydration is often evaluated using impedance measurements based on capacitance or conductance, but we saw above that it can also be estimated through the concentrations of electrolytes in sweat. The possibility of developing non-invasive monitoring devices is therefore of paramount importance. Berardesca et al. recently published a review on the topic of in-vivo measurement of skin hydration [66] using non-wearable devices, which are mostly based on conventional electrical impedance methods and open-chamber evaporimeters. We will not review these devices here and rather focus on wearable ones. 

Kano et al. [67] developed a fast-response and flexible conductance humidity sensor for real-time monitoring of water evaporation on skin. For that, a silicon-nanocrystal film was deposited from a colloidal solution by spin-coating on a PI substrate. The device was extremely sensitive: the current flowing through the Si nanocrystal film varied by 5 orders of magnitude in a relative humidity range of 8–83%, with a response/recovery time of 40 ms (Figure 21). It should be noted that the device was not evaluated when worn on skin; conversely, for measurements, a finger (for example) was brought close-by the bench-top device.

Trung et al. [68] proposed in 2017 a stretchable and conformal sensor for continuously monitoring the moisture level of skin. The sensing layer was made of a blend of reduced graphene oxide and polyurethane, deposited onto PEDOT:PSS electrodes and supported on a PDMS substrate. For real situation measurements, the authors attached their device to a finger for monitoring the humidity level of the skin. The response and relaxation time are 3.5 and 7 s, respectively. Resistance did not depend on stretching up to a strain of 60% and even after 10,000 stretching cycles at 40% strain. Also in 2017, Kabiri et al. [69] described a tattoo-like skin sensor with sub-micrometer thick (total thickness of 460 nm) graphene electrodes having the shape of serpentines, and presenting a stretchability of ca. 40% without loss of conductivity. The electrodes were directly tattooed on skin. With a commercial liquid bandage added on top, this tattoo stayed functional for several days. Humidity was measured by electrical impedance between two tattooed electrodes. Values were consistent with medically used silver/silver-chloride (Ag/AgCl) gel electrodes, but tattoos obviously offered superior comfort (Figure 22).

Yao et al. (2017) [70] described a wearable, fully conformable, skin hydration sensor based on overall impedance. They used two interdigitated silver electrodes (more precisely, the Ag ink was made of silver nanowires, AgNWs) embedded into a PDMS matrix. The hydration sensor was calibrated against a commercial skin hydration system and was included into a flexible wristband, together with a miniaturized network analyzer chip, a small battery, and a Bluetooth chip for data transmission. A chest patch was also developed on the same basis (Figure 23). Even if designed for continuous monitoring directly on skin, the authors did not perform long-term experiments to discuss the stability of their sensor under operation.

### 3.3. Monitoring of Skin Wounds

Skin wound healing is quite a complex phenomenon consisting of a large number of different phases strictly organized according to a highly sophisticated temporal sequence. Although many aspects of this process still remain open questions, medical scientists have identified some steps that are crucial to speed up healing time, reduce scarring and restore skin functions [71]; this is particularly important for those wounds contaminated by pathogens as the healing process may be significantly altered by the infection and the presence of necrotic tissue. Sensors for skin wound healing monitoring may be roughly divided into two categories: i) sensors monitoring physical parameters of the wound (pressure applied to the wounded area, wound depth and temperature); ii) sensors detecting biochemical parameters of the wound (mostly measuring the concentration of enzymes involved in tissue degradation and granulation but pH is another parameter to be taken into account).

A recent example of a sensing platform belonging to the first category was presented in 2017 by McNeil et al. [72]. This paper described a wearable wireless sensor system for continuous monitoring of skin pressure, temperature, and relative humidity. The system was composed of a thin (<1 mm) waterproof adhesive patch on which the commercial silicon-based sensors, microcontroller, wireless transmission antenna, and a polymer battery were mounted. The system can be attached to the patient’s skin with minimal discomfort and can provide reliable measurements up to seven days. It is ideal to monitor the evolution or to prevent the formation of pressure ulcers. A different approach, based on simpler, chipless printed temperature sensors based on thin-film resistors made of PEDOT:PSS and fabricated on flexible, plastic substrates was proposed by both Yamamoto et al. [73] and Nakata et al. [64]. Messaoud et al. [74] reported a platform for moisture monitoring in wound dressings. The sensor was composed of an array of 14 flexible electrodes, screen-printed on a flexible support and placed on the top of a wound dressing. This sensor was able to map moisture levels at the wound interface, which offers perspectives for determining the optimal dressing change without need for visual inspection associated with risk of damaging the periwound skin. Experiments were performed on a model system but not on real wounds. Mehmood et al. [75] also reported a portable system for monitoring moisture level inside a dressing. The real-time data were transmitted wirelessly to a portable receiver which displayed the measured values. Measurements were validated on healthy human legs, using commercial compression bandages and dressings (Figure 24).

One should underline the fact that commercial products already exist. For example, WoundSense [76] is a commercial device for measuring dressing humidity. It is a single-use, sterile, moisture sensor to be placed in dressings, working with a hand-held electronic (non-wearable) reader, plugged onto the sensor when a reading is needed. WoundSense can be used in hospitals or at home and allow better planning for dressing changes. WoundSense received the CE mark. Milne et al. [77] published a study using the WoundSense moisture sensor which was put directly inside dressings and was able to provide the moisture status of the wound without need for removing the dressing. The results of the study showed that roughly 50% of the dressing changes are unnecessary because they are made while the moisture level is ideal. This large number of unnecessary dressing changes could be avoided if smart dressings were systematically used (Figure 25).

Rahimi et al. [78] described a flexible array of pH-sensitive electrodes fabricated on a polymer-coated paper. Electrodes were screen-printed with a carbon ink then coated with a conductive proton-selective polymer (polyaniline). Reference electrodes were conventional screen-printed Ag/AgCl ones. The sensitivity is consistent with the state-of-the-art, and the response time is between 10 and 40 s. The sensor biocompatibility was confirmed by cultivating human kertinocyte cells on top of the electrodes, but no in-vivo on-skin measurements were performed, or integration into a dressing or a bandage. McLister et al. [79] developed an original electrochemical strategy for measuring wound pH. Their redox probe is a solid peptide homopolymer of tryptophan, in the form of wires, which gives quinone-containing derivatives after a first irreversible oxidation step (Figure 26). Again, no experiments were performed in-situ, but the sensitivity of the electrode was assessed with horse blood, with a corresponding pH measured at 7.5 ± 0.1.

Still on pH or humidity sensing, Bhushan et al. [80] published in 2019 an enzymatic biosensor for monitoring uric acid in wound exudates. The sensing pathway was conventional, with amperometric determination of uric acid (UA) through oxidation by urate oxidase and horseradish peroxidase. The lowest detectable uric acid concentration was ca. 10 μM. Experiments were done only ex-situ, however. RoyChoudhury et al. [81] also reported an enzymatic biosensor for uric acid quantification in wound fluid (Figure 27). Urate oxidase (UOx) was entrapped into a polyvinyl alcohol-based cationic polymer and ferrocene carboxylic acid was used as redox mediator. The sensor showed a stable response at body temperature for a week. The quantity of necessary wound fluid was of few μL only. UA was measured repeatedly at levels between 1 and 50 μM. A potentiostat and a microcontroller were integrated on a flexible polyimide substrate, for future integration as a wearable device, but no experiments have been conducted in-situ, directly on a patient’s wound.

Perhaps the most complete example of physical sensing system for skin wound evaluation and monitoring published so far was the one presented in 2017 by Chang et al. [82]. The system was composed of a wound assessment probe equipped with a light source, an optical RGB camera, a depth camera, a thermal camera, and an array of chemical sensors. This probe was used to scan the wound’s surface and transmitted the data to a computer connected to a monitor visualizing the results thanks to a dedicated user interface. The system was capable of performing the following analyses: 1) tissue segmentation and classification; 2) 3D modeling for wound measurements; 3) thermal analysis; 4) multi-spectral analysis (i.e., detection and comparison of hemoglobin O_2_ saturation in both the wound tissue and in the healthy surroundings); 5) chemical sensing (i.e., analysis of the vapors emitted by the wound). Bruinink et al. [83] recently published a review on skin wound management. Contrary to physical sensors, biochemical sensors for cutaneous wound healing monitoring are essentially sensors capable of detecting and quantifying target molecules present in the exudate produced by the skin tissues in the first phases of wound formation and healing. Not many examples of such type of sensors have been reported so far in the literature; one should notice, however, that since some of the molecules of interest (urea, uric acid, ions, etc.) are the same molecules found in the sweat produced by healthy skin, one could imagine using the sensors developed for sweat analysis and described in the previous parts of this review also to evaluate and monitor a wound, even if the fluid collection system has to be reengineered [84]. Most biochemical sensors are nowadays focused on the detection of C-reactive protein (CRP), a protein able to promote the action of macrophages to remove necrotic and apoptotic cells and bacteria. By far, the most common detection method is represented by immunoassays and, in particular, by enzyme linked immunosorbent assays (ELISA) even though some alternative methods, based on field-effect transistors [85], surface plasmon resonance [86], and electrochemical impedance spectroscopy (EIS) [87] have been used. Still, to the best of our knowledge, all the examples reported in the literature require that blood or interstitial fluid samples be taken from the patient’s body; an example of a sensor that can be applied directly on skin is still missing. Continuous monitoring of wound recovery is vital, however, for preventing chronic ulcers; this is particularly true for patients with diabetes. Common practices require patients’ co-operation, frequent bandage changes and regular blood analysis conjugated with time consuming laboratory assays, which discourages patients from seeking proper medical attention and, in parallel, motivates researchers and engineers to look for wearable point-of-care biosensors integrated into bandages and dressings, combining several conformable chemical sensors on a single device [88]. This is the reason why future smart dressings need integration of sensors which can diagnose wound chemical biomarkers such as oxygen level, pH, lactate, glucose and interleukin-6 (IL-6) (Figure 28).

### 3.4. Alcohol and Drugs Detection

Alcohol abuse has multiple consequences such as serious health problems for heavy drinkers, violent behavior and car crashes, leading to considerable socioeconomic costs. Various methods are already used for rapid quantification of alcohol consumption, based on measurements in breath, urine, and blood. Blood alcohol concentration is, for the moment, the most precise data and is the most commonly used indicator but is obviously the least convenient, which limits its use to serious cases and limits its use in prevention of alcohol-related accidents. Therefore, there is still a demand for easier, but reliable, methods for assaying blood alcohol. The most common alternative is the breath analyzer, but it is not fully reliable if the environment contains interfering chemicals. This is the reason why apparatuses able to measure transdermal alcohol concentration (TAC) have been recently developed. As soon as 1999, Buono et al. [90] reported the comparison of blood ethanol concentration (BAC) and TAC levels, through a study on 10 subjects, using a non-wearable sensor. They reported a correlation of 0.98 between sweat and blood ethanol, with a slope of 0.81 between TAC vs. BAC (which means that TAC is lower than BAC). These results demonstrated that ethanol can be detected in sweat and paved the way for further technical developments. In this review, we will not discuss the non-wearable alcohol sensors, which have been extensively described and work routinely, but we will focus on wearable technologies.

In 2014, Gamella et al. [91] also described a non-invasive sensing device for indirect estimation of BAC by monitoring transdermal alcohol concentration (Figure 29). The detection principle was based on a conventional bienzyme amperometric mechanism. The prototype, which was small but relatively thick compared to a tattoo-like skin patch, was put in direct contact with skin on 40 subjects and the results compared to BAC and breath alcohol concentration (BrAC) values for each; the device showed a better sensitivity than the breathalyzers on the market at the time, and the response time range was acceptable (within 5 min). However, the measurement required an initial stimulation of sweating (5 min), a mandatory step not consistent with real continuous measurements.

Kim et al. [92] reported in 2016 a tattoo-like sensor for alcohol monitoring in sweat, which was evaluated directly on skin under real conditions. It integrated an iontophoretic part, an amperometric sensing part and flexible wireless electronics (Figure 30). Iontophoresis was not used to get interstitial fluid, but to deliver pilocarpine which induced sweat production; in turn, ethanol was detected using a conventional alcohol-oxidase/Prussian blue electrode. On-body experiments showed an increase of ethanol levels after alcohol intake. The integrated electronic board was able to control the iontophoresis/amperometry operation and transmit data wirelessly via Bluetooth. However, the paper did not provide any indication on the possibility of using this tattoo continuously and above all, there is still a potential for improvement to achieve a real integrated tool. Indeed, here, the flexible wireless instrumentation electronics is not included in the tattoo.

Lansdorp et al. (Milo Sensors, Inc.) reported very recently a wearable transdermal alcohol sensor [93] in the form of a wristband, which allows measurements up to 24 h in a row. The detection principle is similar to the one of the previous examples (alcohol oxidase + Prussian blue); it delivers a linear response between 0 and 50 mM. The authors showed continuous data recorded with their device on a subject for two consecutive days, with accurate results (Figure 31). The sensor is composed of a disposable capsule made of an electrochemical device with three electrodes covered with a hydrophilic membrane in contact with the skin. In this remarkable example of integration, transdermal alcohol is recovered passively without the need for stimulation.

Lawson et al. [94] developed a SnO_2_ sensor for monitoring ethanol on the skin. Transdermal alcohol emissions by perspiration were investigated during clinical trials on six subjects to demonstrate the relevance of this method. They also observed that skin emissions of ethanol are consistent with blood and breath concentrations (Figure 32). This type of sensor represents an interesting alternative to the use of sweat as an analyte vector by circumventing the problems of residue accumulation or the need to stimulate perspiration.

Apart from alcohol, it is also possible to detect consumption of drugs using skin sensors. Reports concerning these applications are scarce, however. Voss et al. [95] described in 2014 a portable (even if not wearable) electronic nose to detect directly on skin some changes in body odor caused by cannabis consumption. The authors enrolled 20 cannabis-smoking subjects for this study and compared their results to 20 tobacco-smoking ones. Because signals were difficult to interpret on a simple basis, two data processing techniques were used: principle component analysis (PCA) and pattern recognition with subsequent support vector machines (SVM). The SVM analysis obtained an accuracy of 92.5% between cannabis-consuming volunteers and tobacco-smoking subjects. Technically, the electronic nose consisted of relatively non-specific gas sensors, one suitable for combustible gases, one for hydrocarbons, and the third one for NO_2_. More recently, Mishra et al. [96] developed a wearable potentiometric tattoo biosensor for monitoring various toxins such as organophosphates. Diisopropyl fluorophosphate (DFP) was used as a model. It should be noted, however, that even if this sensor was directly put on skin, it was not intended to measure skin secretions: it was rather used to detect the presence of warfare chemicals in the direct environment (liquid or gas). The detection principle relied on the use of polyaniline as pH-sensitive layer, to monitor the enzymatic hydrolysis of DFP by the organophosphate hydrolase (OPH) enzyme (Figure 33A). The electrodes were screen-printed on a tattoo paper (Figure 33B,C) and then connected to a flexible electronic interface for wireless data transmission.

### 3.5. General Skin Status, Reactive Oxygen Species, and Skin Microbiota

The role of skin microbiota—e.g., the bacteria which are normally present on skin—is being now widely studied, because changes of microbial population (i.e., change in the ratio of a bacterial species within a microbial population) often translates more profound pathologies such as psoriasis, atopic dermatitis, or other skin diseases. Often, the bacterial equilibrium is broken by an overabundance of *Staphylococcus aureus*. Conversely, if the number of *Propionibacterium acnes* (*P. acnes*) bacteria goes below a critical threshold, a predisposition for developing skin diseases based on oxidative stress may develop. This is the reason why some groups developed sensing approaches, some being wearable, to try to characterize skin health. This has been done either by characterizing skin porosity, skin enzymatic activity, or bacterial activity.

As soon as 2012, Farrow et al. [97] described a system to characterize bacteria on skin, more particularly in wounds. Their detection system was based on an Ag/AgCl electrode used for electrical impedance measurements when placed into cultures of various bacteria. However, even if this device was able to detect, unspecifically, high concentrations of bacteria, the presence of Ag^+^ ions from the electrode inhibited bacterial growth, which biased the analysis conclusions. This approach has not been further investigated in the more recent literature. In 2017, De Guzman et al. developed a screen-printed tattoo sensor for the assessment of the skin barrier integrity. The tattoo comprised two concentric flexible circle electrodes used to perform impedance spectroscopy directly at the level of the outer stratum corneum (SC) (Figure 34). The SC is known indeed to play a critical role in the barrier function of the skin (e.g., protecting tissue from infections, dehydration, or even chemicals). Data obtained from this sensor were compared to ‘tissue dielectric constant’ (TDC) measurements obtained from the commercially available MoistureMeterD (MMD, Delfin Technologies). The tattoo sensor was able to reliably identify changes related to skin hydration/dehydration and was proposed as a technical aid for the management of skin diseases, such as atopic dermatitis or psoriasis.

More recently, the same team [99] described a thin, flexible sensing platform based on silver electrodes screen-printed on a thin elastomeric substrate for skin monitoring. Using an acrylate porous adhesive layer improved the mechanical properties of the patch without degradation of the electrical performances. For measurements, a conventional table-top impedance setup was used (Figure 35). Composed mainly of Ag and plastic materials, this device is a remarkable example of disposable sensor composed of relatively abundant, low-toxicity materials (e.g., silver, ethyl cellulose). The use of such materials is facilitated by the poor degree of integration of the sensor with a measurement made from an external and dedicated device using wired connection.

Nocchi et al. [100] investigated the possibility to characterize the well-being of skin through the activity of the catalase enzyme which is naturally present in skin, as a part of the biological antioxidative system. For this purpose, a conventional oxygen electrode was covered with a viable pig skin (Figure 36A), and various concentrations of hydrogen peroxide were delivered. As shown in Figure 36B, H_2_O_2_ is able to penetrate the stratum corneum (SC), diffuse through the underlayers and then react with catalase to give O_2_, which in turn diffuses back to the electrode. The authors have shown that removing partly the SC layer (e.g., by a mechanical effect or by repetitive tape-stripping) results in a 10-fold increase of the current—i.e., H_2_0_2_ diffusion to skin. In short, this study has shown that the state of skin can be assessed by evaluation of the catalase activity. What is now necessary is the development of a sensor that can non-invasively characterize this catalase activity, which has not been reported yet.

Following a totally different strategy, Bergdahl et al. [101] reported in 2019 a surface plasmon resonance sensor for the quantification of a bacterial factor (RoxP) secreted by skin. RoxP is indeed an indirect protein biomarker of skin health produced by *Propionibacterium acnes*; a lack of RoxP indicates a predisposition for developing skin diseases based on oxidative stress. The sensing layer was a molecularly imprinted polymer (MIP) and transduction was performed by SPR. The limit of detection (LOD) value, obtained on real samples (skin swab) was ca. 0.25 nM—i.e., better than common ELISA tests. Carmona et al. [102] used another approach for characterizing the presence of bacteria in sweat: they used an electronic nose made of two SnO_2_-based gas sensors (nanowire and thin film) integrated together to deliver a global signal further treated by principal component analysis. Different mixtures of microorganisms were grown in artificial sweat and the results from the developed sensor were compared to conventional GC-MS (with solid phase microextraction). Discrimination between various types of microbiota was effective. The sensor was not wearable, however.

### 3.6. Patents

Because skin sensors have raised a considerable interest these last years, a relatively large number of patents have been submitted in this field. Below are given some pertinent examples. Except one, they all use the same principles discussed above.

Schroers et al. [103] proposed a basic conductimetric device able to detect moisture on skin. The electrodes are directly deposited on a textile patch (resistant to stretching, bending or compression by nature). The original application targeted by this invention is to monitor fluid leakage at the puncture point of a patient’s arm during transfusion or any other extracorporeal blood treatment. Cavallari et al. [104] claimed the basic idea of interconnected tattoos. They described a conformal tattoo biosensor including a pattern of sensors made of a conductive polymer, electrically connected to a contact region and to a wearable signal monitor able to transmit signals wirelessly. Ziaie et al. [105] claimed a colorimetric wearable paper-based sensor for monitoring perspiration, made of two components: a disposable sweat collection patch carrying visual indicators (Figure 37A) and a reusable electronic module for reading and data transmission, which alerts the user when the film collects a pre-determined volume of sweat. This patch is designed to help athletes to assess their sweat loss. Lansdorp et al. [106] proposed a wearable device, in the form of a wristband, able to measure an analyte in sweat. Their patch includes a sensing selective membrane, a reservoir to receive the target analyte, and an enzyme-modified working electrode able to deliver a current proportional to the analyte to measure. Its functioning is illustrated for ethanol quantification in sweat, using alcohol oxidase and a conventional amperometric setup with HRP and ferrocene as secondary enzyme and mediator, respectively. The wristband (Figure 37B) also includes a wireless transmitter.

At last, Bangera et al. [107] reported a protocol for assessing the microbiota of skin. They proposed a mask made of a material that is put in contact with skin. The inner surface of this mask contains a distribution of colorimetric sensing area (the inventors propose ELISA-based colorimetric sensors). After a given contact time, the inner surface of the mask is analyzed with a camera which is able to map the microbiota as a function of the skin areas (Figure 38). It should be noted here that several practical or scientific difficulties have not been addressed, but the overall idea is interesting to keep in mind.

## 4. Conclusions and Perspectives

Through the various developments which are reviewed herein, the reader understands that wearable chemical skin sensors are more demanding than conventional sensors in terms of technologies, even if most transduction schemes repeat the same pathways as most ions or enzymatic sensors originally designed to work in solution (polluted waters, urine, interstitial fluid, etc.) and are not innovative on this particular aspect.

The achievement of wearable analytical tools requires addressing two distinct issues. The first one is the sampling method. Indeed, apart from the devices using micro-needles allowing a direct sampling of the interstitial fluid, many approaches are based on the analysis of sweat. In this case, sweat collection is often overlooked: how is sweat collected? Is there any mixing with previously emitted sweat that could lead to sensor hysteresis? Currently, this technological challenge is only partially won.

The other major problem concerns the integration of the sensing parts within a complete wearable device, which can be described in terms of six main issues.

First of all, the substrate—i.e., the part of the device which will carry the active layers—has to be made of a highly conformable material and be as thin as possible. On this aspect, tattoos are certainly the most promising form of skin sensors, being extremely light and as flexible as the skin itself; they are minimally invasive devices. The other very promising devices are those directly made on textile, because they can be directly integrated in our garments; from this point of view, they are the least invasive devices.

Second, patches, tattoos, or textiles materials must resist to some aggressive fabrication steps such as photolithographic processes and associated chemicals, and/or thermal annealing. From this point of view, some improvements have to be done because most chemically or thermally resistant materials are not flexible enough to provide the wearer with sufficient comfort, an exception being represented by silicone-based materials.

Third, the active layers must be as flexible as the substrate without loss of performance. This is equally true for colorimetric and for electrochemical devices, even if on this aspect the latter technology is intrinsically more fragile; conversely, its performances are usually better so that compromises have to be made depending on the particular application. Electrochemical devices need conductive tracks and working electrodes. Such parts can be advantageously printed, through conventional screen-printing (many examples cited in this review use screen-printing, but electrical performances, thicknesses, and lateral resolutions can be improved using the emerging printing technologies such as inkjet-printing; few skin sensors use this technology yet, which is probably the most efficient, however).

Fourth, electrochemical devices have to be connected to signal processing units and wireless transmitters which have to be integrated on flexible substrates even if they are not flexible themselves. This hybridization between flexible and rigid electronics induces many difficulties in terms of electrical contacts; one way to solve this problem is to be capable of making such processing and transmission units flexible as well, therefore made from organic (printed) electronics. This is a rapidly growing field of research, which can also be applied to the sensing element as well (e.g., by replacing conventional amperometric electrodes by printed transistors). A very recent example of this technology was given, for example, by Li et al., 2019 [108].

Fifth, active skin sensors rely on a power source. Even if some batteries are flexible and even printed, energy harvesting through thermoelectric or piezoelectric transducer seems more pertinent for low-power sensors.

Last, the price of these sensors must be extremely low and based on ecofriendly materials as they are, by definition, more easily disposable. Again, this tends to lead fabrication technologies toward printed electronics instead of conventional microlithography.

## Figures and Tables

**Figure 1 sensors-19-04376-f001:**
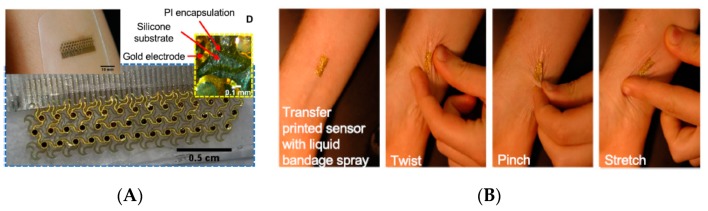
(**A**) Image of the gold electrodes array on a polymeric wound dressing. (**B**) Pictures of a transferred printed flexible array, fixed with a liquid bandage, under mechanical stress. Reproduced from [27] with permission. Copyright 2019 Elsevier B.V.

**Figure 2 sensors-19-04376-f002:**
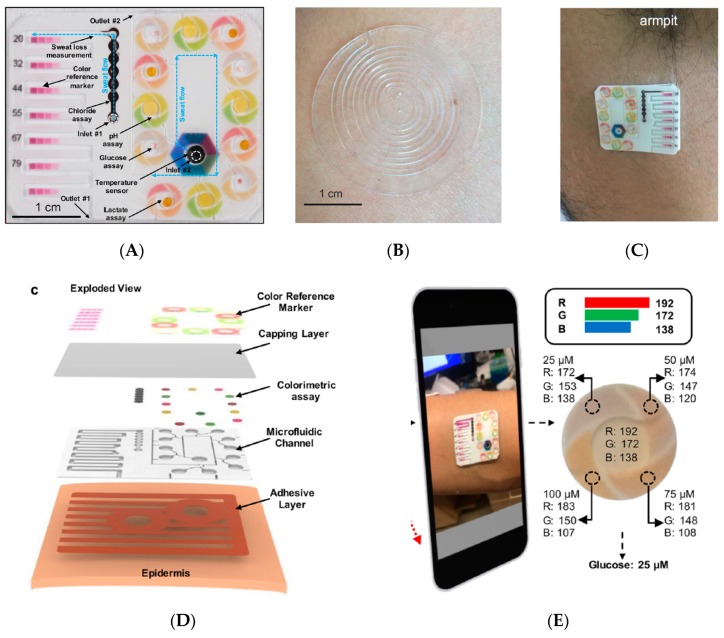
(**A**) PMMA patch onto which the microfluidic channels and reservoir are engraved. The printed patterns are used as visual references. (**B**) Soft PDMS patch for collecting sweat. The width and depth of the coiled channel are 1 mm and 300 μm, respectively. (**C**) Patch placed close to the armpit. Some sweat (internally mixed with a dark dye) started to flow. (**D**) Exploded view of the patch. (**E**) Example of data transmission by image treatment using a smartphone. Reproduced from [28] with permission. Copyright © 2019, American Chemical Society.

**Figure 3 sensors-19-04376-f003:**
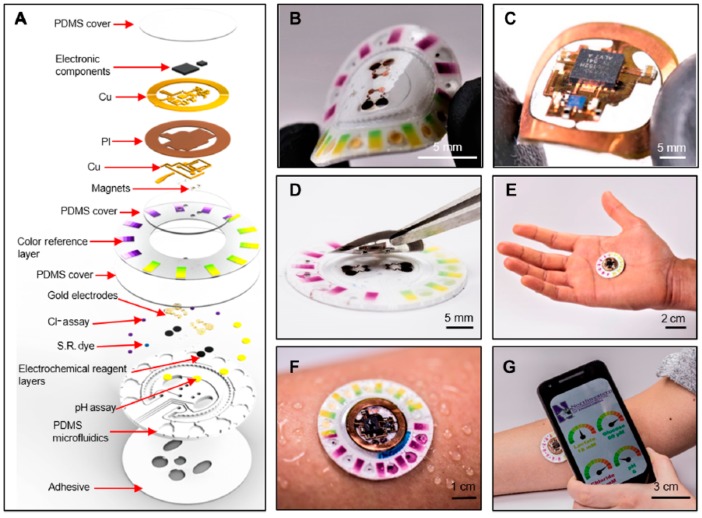
(**A**) Exploded view of the colorimetric/amperometric device. PI: polyimide. S.R.: sweat rate. (**B**) Colorimetric and microfluidic parts of the patch upon bending and (**C**) NFC electronics (**D**) that are placed on top of the patch. (**E**,**F**) Complete device into a hand and fixed on a forearm. (**G**) Illustration of a smartphone application for wireless communication with the patch. Reproduced from [30]. Copyright © 2019 The Authors, some rights reserved. Distributed under a Creative Commons Attribution Non-Commercial License 4.0 (CC BY-NC).

**Figure 4 sensors-19-04376-f004:**
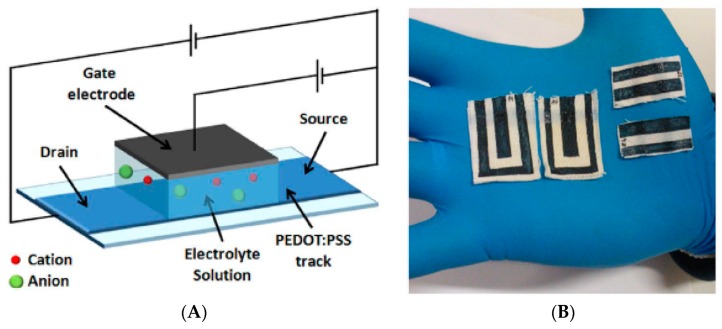
(**A**) General scheme of an OECT. All components are in PEDOT:PSS. (**B**) Screen-printed OECTs (devices on the left, fabricated on woven cotton (250 μm thick) and those on the right printed on a Lycra substrate). Reproduced from [31]. Creative Commons Attribution 4.0 International License. Copyright 2016 Springer Nature Publishing AG.

**Figure 5 sensors-19-04376-f005:**
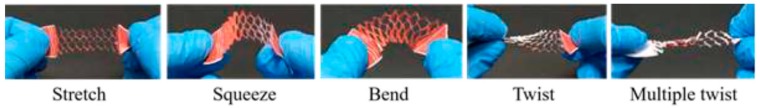
Patterned paper fish under various mechanical stresses. Reproduced from [32] with permission. Copyright 2018 WILEY-VCH Verlag GmbH & Co. KGaA, Weinheim.

**Figure 6 sensors-19-04376-f006:**
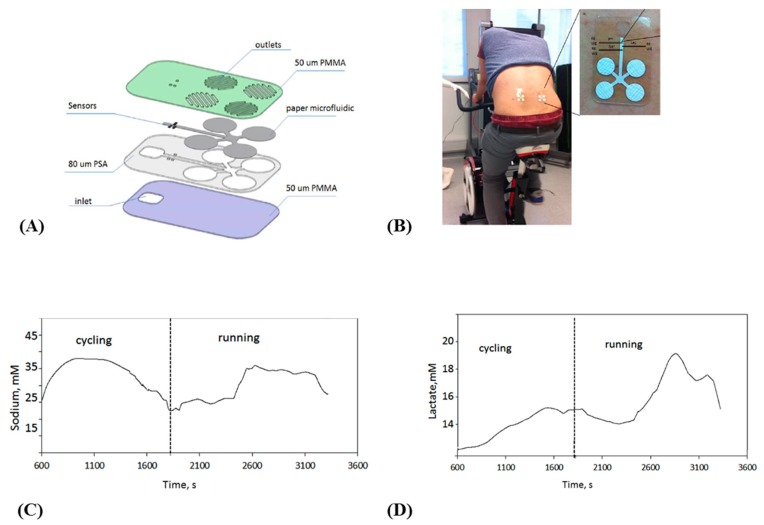
(**A**) Scheme of the four layers constituting the microfluidic chip (PSA: pressure sensitive adhesive). (**B**) Place where the patch is attached to the subject during an effort; inset: detail of the patch. (**C**) Measured lactate and (**D**) sodium cations in sweat during an effort. Adapted from [33]. Creative Commons Attribution License (CC BY), 2017.

**Figure 7 sensors-19-04376-f007:**
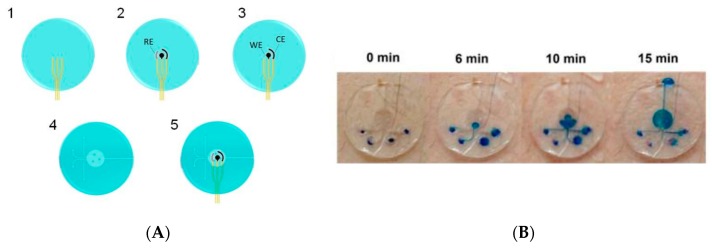
(**A**) Microfluidic electrochemical patch with (1) Gold current collectors lithographied then transferred on the PDMS substrate; (2,3) Reference (RE) screen-printed with silver/silver chloride (Ag/AgCl), working (WE) and counter (CE) electrodes screen-printed with Prussian blue; (4) PDMS microfluidic layer finally bonded (5) on top of the PDMS electrode layer. (**B**) View of the complete patch glued on skin, as a function of time during effort. Sweat is mixed with a blue dye inside the device for this picture. Reproduced from [34] with permission. Copyright © 2017, American Chemical Society.

**Figure 8 sensors-19-04376-f008:**
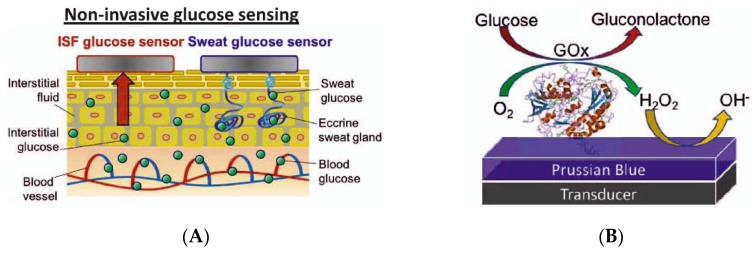
(**A**) Depiction of two different wearable epidermal glucose sensors using the interstitial fluid (ISF) or sweat. (**B**) Basic functioning of the enzymatic amperometric glucose sensor. Reproduced from [35] with permission. © 2017 Elsevier B.V. All rights reserved. (**C**) Glucose sensing in ISF through reverse iontophoresis (bottom-right) and GlucoWatch^®^ biographer display. Reproduced from [36] with permission. Copyright © 2002 John Wiley & Sons, Ltd. Left: reverse iontophoresis process. Reproduced from [37] with permission. Copyright © 2001 Elsevier Science B.V. All rights reserved. (**D**) Top: Picture of a flexible sweat extraction and sensing device along with the data-treatment electronics on a flexible board. Bottom: iontophoresis and sensing mode of operation. Reproduced from [38] with permission. Copyright 2017, National Academy of Sciences.

**Figure 9 sensors-19-04376-f009:**
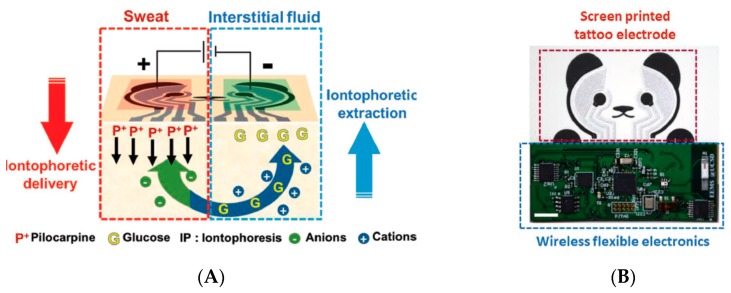
(**A**) Scheme of iontophoretic delivery of pilocarpine and iontophoretic extraction of glucose. (**B**) Screen-printed biosensor coupled with wireless flexible printed circuit board. The left-hand site is dedicated to alcohol sensing while the right-hand side is dedicated to glucose. White scale bar: 7 mm. (**C**) Demonstration of the flexibility of the overall device (board + sensors). (**D**) Tattooing the sensor part. (**E**) Sensing performance after meal then alcohol intake (left) or alcohol then meal intake, compared with blood glucose and breath alcohol. Reproduced from [39]. © 2018 the authors. Published by WILEY-VCH Verlag GmbH & Co. KGaA, Weinheim. https://creativecommons.org/licenses/by/4.0/.

**Figure 10 sensors-19-04376-f010:**
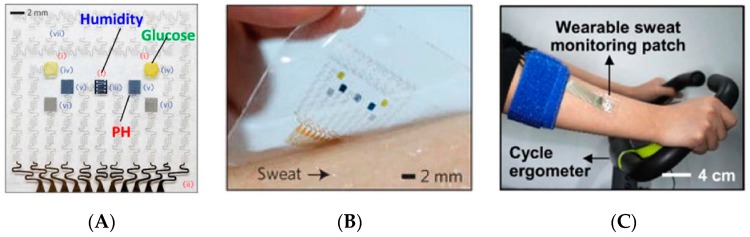
(**A**,**B**) Flexible patch and its transfer on skin. (**C**) Sweat generation with the wearable patch on the subject’s arm. Reproduced from [40]. Copyright © 2017, The Authors. Creative Commons Attribution-NonCommercial license. (**D**) Comparison between the patch, a glucose strip, and continuous glucose monitoring (CGM) over the course of a day. Reproduced from [41] with permission. Copyright © 2016, Springer Nature.

**Figure 11 sensors-19-04376-f011:**
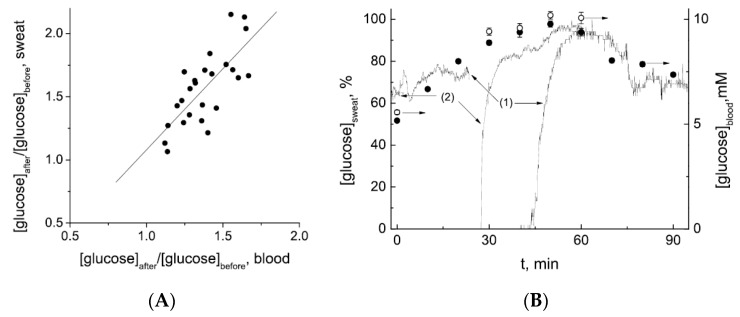
(**A**) Correlation in ratios of glucose content after glucose loading to the corresponding value before loading, for sweat and blood. Pearson coefficient r = 0.75. (**B**) Normalized sweat glucose level (solid lines) in comparison with blood vein glucose (circles) during a glucose tolerance test. The black and white circles correspond to two different measures. Curve 2 was obtained on a patient whose sweat has not been stimulated beforehand (sweat stimulation took about 25 min), while curve 1 was obtained on a patient whose sweat has been stimulated well before the glucose test. The breaks in the curves correspond to a second sweat stimulation. Reproduced from [45] with permission. Copyright © 2019 American Chemical Society.

**Figure 12 sensors-19-04376-f012:**
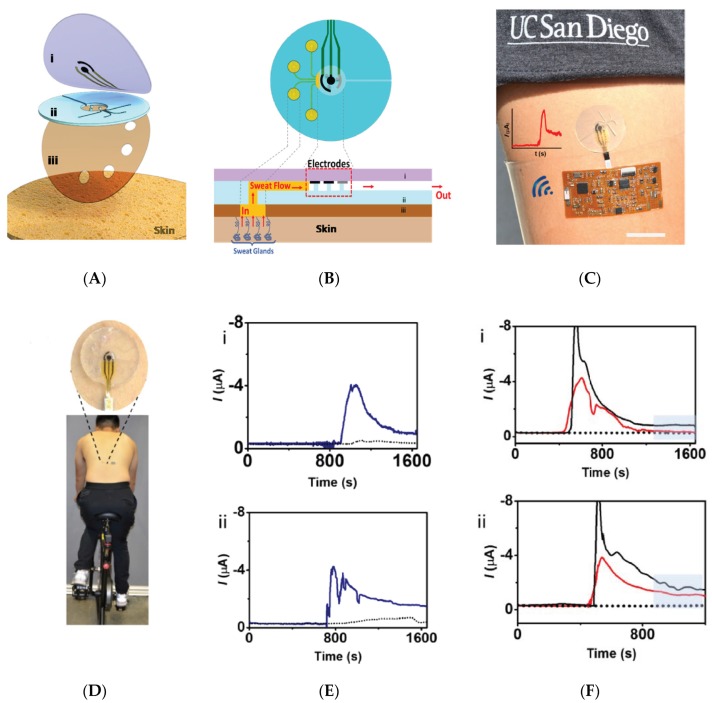
(**A**) Scheme of the three-layer device: (i) top PDMS layer carrying sensing electrodes; (ii) middle PDMS layer carrying microfluidics; (iii) adhesive layer in contact with the skin. (**B**) Sweat collection. (**C**) Device integrated with wireless flexible electronics. Scale bar: 5 mm. (**D**) Position of the sensor on the back of the subject during effort. (**E**) Continuous lactate monitoring with (blue full line) and without (dotted line) lactate oxidase for (i) subject 1 and (ii) subject 2. (**F**) Continuous glucose monitoring before (red full line) and after meal (black full line) with glucose oxidase and without (dotted line) for (i) subject 1 and (ii) subject 2. Amperometric experiments were carried out at −0.1 V vs. Ag/AgCl during physical exercise, and all data were wirelessly transmitted to a computer. Reproduced from [34] with permission. Copyright © 2017 American Chemical Society.

**Figure 13 sensors-19-04376-f013:**
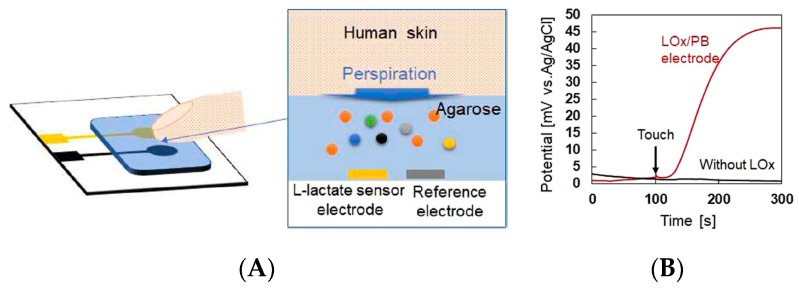
(**A**) View of the agarose-based lactate sensor. (**B**) Potentiometric response of the lactate oxidase modified sensor upon subject’s forefinger contact. Reproduced from [48]. http://creativecommons.org/licenses/by/4.0/.

**Figure 14 sensors-19-04376-f014:**
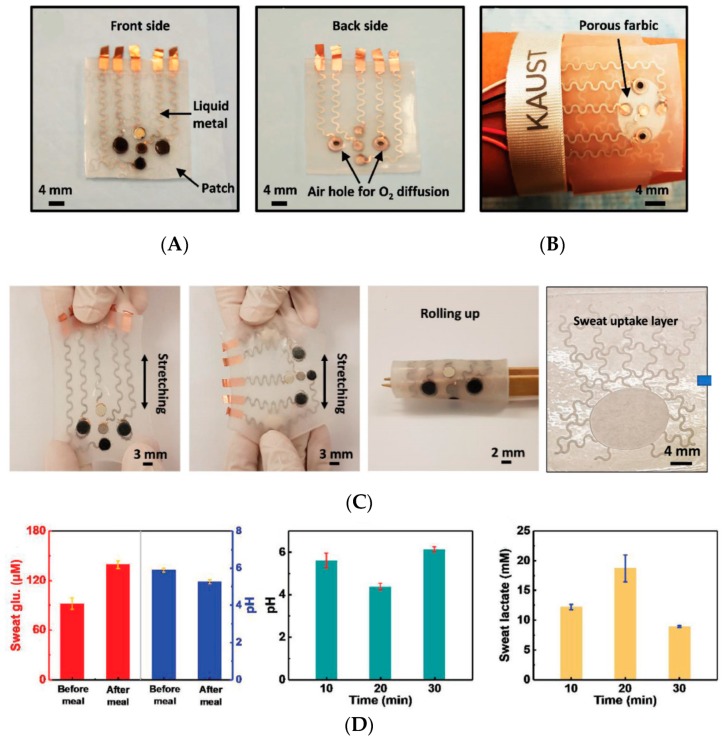
(**A**) Picture of the front and back sides (left, middle) of the sensor array with contacts, conductive tracks and electrodes (pH, lactate, and glucose). Right: Sensor wristband on subject’s forearm. (**B**) Stretching in both directions and rolling up. (**C**) Patterned underlayer for sweat collection. This layer, which is put in direct contact with the skin, is the backside of the sensors layer. (**D**) Glucose, pH, and lactate monitoring on a subject’s forearm before and after meal, and during effort (maximum effort at 20 min). Reproduced from [54] with permission. © 2019 WILEY-VCH Verlag GmbH & Co. KGaA, Weinheim.

**Figure 15 sensors-19-04376-f015:**
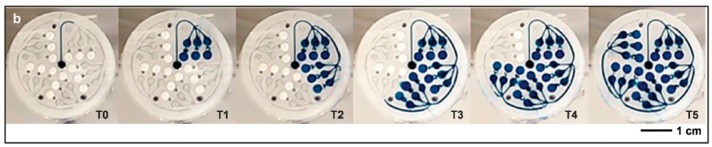
Pictures of the chrono-sampling of sweat (artificially colored in blue), introduced at a rate of 10 μL min^−1^. The different areas are filled sequentially with the colored sweat solution depending on the swollen state of the superabsorbent polymer (SAP) valve. Reproduced from [56] with permission. © 2018 WILEY-VCH Verlag GmbH & Co. KGaA, Weinheim.

**Figure 16 sensors-19-04376-f016:**
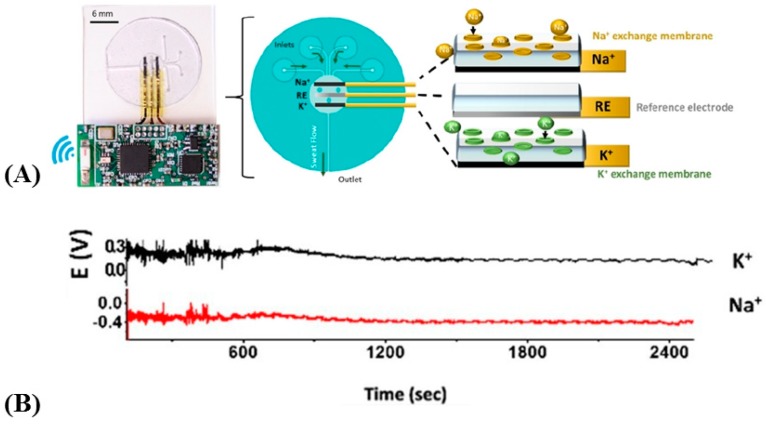
(**A**) Picture of the whole device (left) and schematic view of microfluidic chambers and channels into which are integrated the ion selective electrodes (right). (**B**) Simultaneous recording of the ISE potentials for Na^+^ and K^+^ for a sensor mounted on the shoulder of a bicycle pedaling subject. Reproduced from [58] with permission. © 2019 Wiley-VCH Verlag GmbH & Co. KGaA, Weinheim.

**Figure 17 sensors-19-04376-f017:**
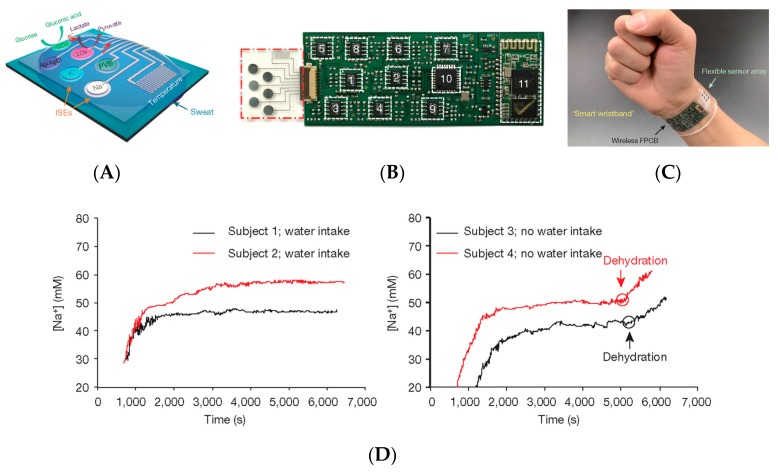
(**A**) Schematic of the sensor array, showing glucose, lactate, sodium, potassium and temperature sensors. (**B**) Photograph of a flattened flexible PCB, with the sensors array on the left, and the 11 electronic components on the right. (**C**) Picture of the active wristlet on a subject’s wrist. (**D**) Real-time sweat sodium and potassium levels during an endurance run with (left) or without (right) water intake. Dehydration is diagnosed in the latter case. Reproduced from [59] with permission. Copyright © 2016, Springer Nature.

**Figure 18 sensors-19-04376-f018:**
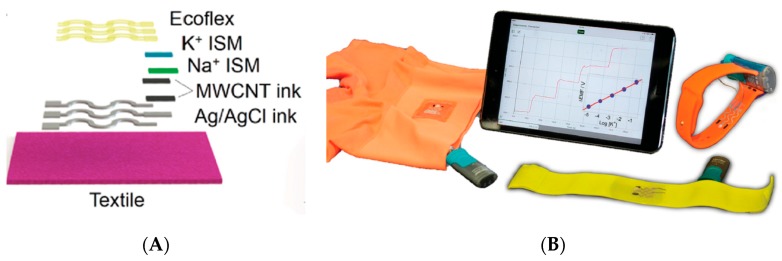
(**A**) Scheme of the manufacturing process on a polyurethane (PU) substrate. (**B**) Examples of use of the sensor, on underwear, wristband, and headband, illustrating the versatility of the printable and stretchable sensor array on different common wearable objects. The tablet displays a real-time trace of increasing potassium levels obtained wirelessly by the underwear printed sensor. Reproduced from [60] with permission. © 2016 WILEY-VCH Verlag GmbH & Co. KGaA, Weinheim.

**Figure 19 sensors-19-04376-f019:**
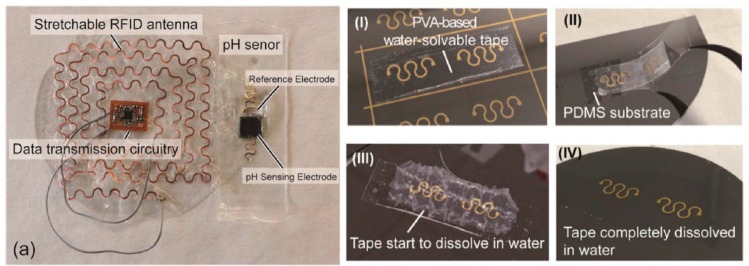
(**a**) Photograph of the stretchable wireless antenna, data transmission circuit and sensor. Right: steps of the transfer protocol of the circuits from Si wafer to PDMS. Reproduced from [63] with permission. © 2018 Elsevier B.V. All rights reserved.

**Figure 20 sensors-19-04376-f020:**
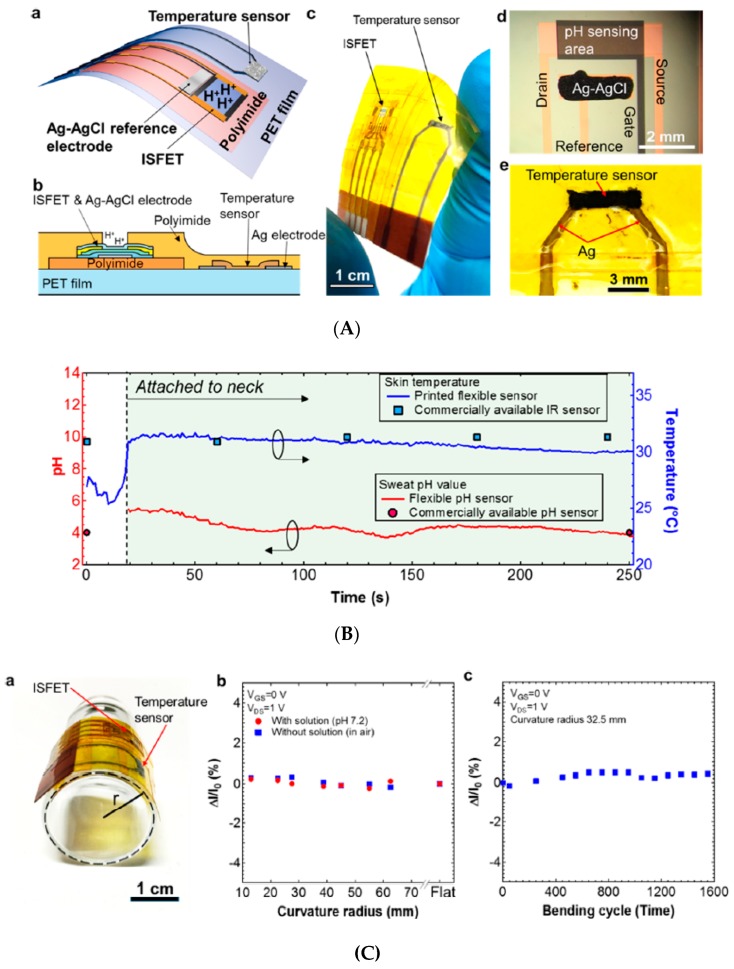
(**A**) a—Scheme of the flexible strip integrating pH and temperature sensors; b—Side view of the encapsulated sensors; c—Picture of the strip held between two fingers. d, e—Details of the two sensors. (**B**) Real-time pH and skin temperature acquired from the neck of a patient, by the device (continuous lines) and measured by commercial pH and infrared sensors (squares and circles). (**C**) a—Picture of the complete strip under bending (r: curvature radius). b, c—Normalized ISFET current as a function of curvature radius and during 1600 bending cycles at a radius of 32.5 mm. Reproduced from [64]. Copyright © 2017, American Chemical Society.

**Figure 21 sensors-19-04376-f021:**
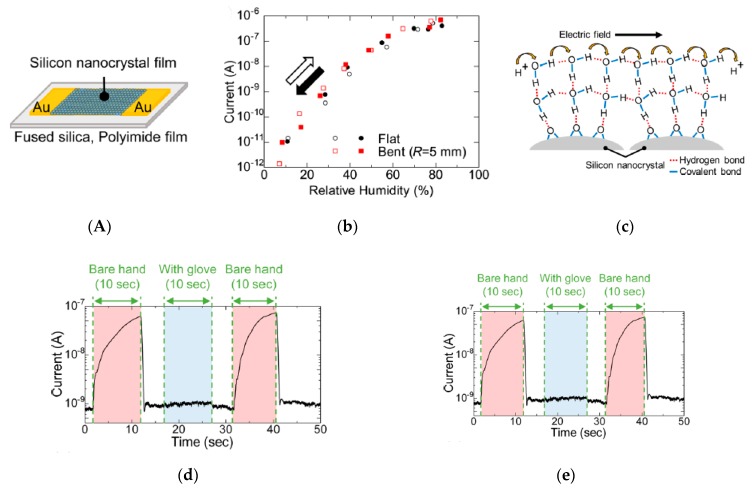
(**A**) Illustration of the Si-NC-based humidity sensor. (**B**) Current of the sensor as a function of relative humidity. Open and filled marks: increasing and decreasing humidity, respectively. (**C**) Illustration of charge carrier transport principle on Si-NC. (**D**) Monitoring of water evaporation from a bare hand skin. (**E**) Real-time detection of water evaporation from a hand. Pink periods correspond to application of a bare hand on the sensor. Reproduced from [67] with permission. Copyright © 2017, American Chemical Society.

**Figure 22 sensors-19-04376-f022:**
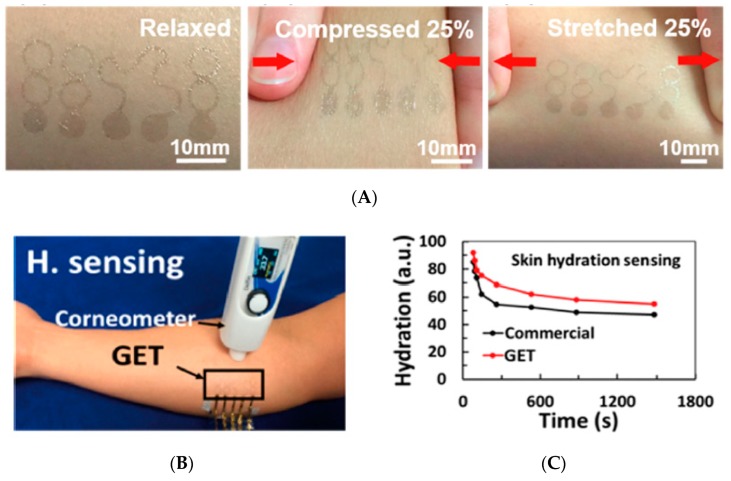
(**A**) Graphene based electronic tattoo mounted on skin. (**B**) Demonstration of the graphene electronic tattoo (GET) sensor as skin hydration sensor, compared to a commercial corneometer. (**C**) Skin hydration after application of body lotion, using the humidity sensor of the GET and the commercial corneometer. Reproduced from [69] with permission. Copyright © 2017, American Chemical Society.

**Figure 23 sensors-19-04376-f023:**
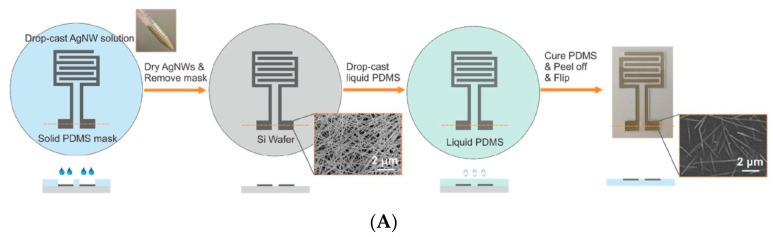
(**A**) Fabrication of the silver nanowires (AgNW)-based sensor, with AgNW network before being embedded into PDMS (left SEM picture) and after embedding (right SEM picture). (**B**) Picture of an AgNW patch placed on the inner side of a forearm. (**C**) Impedance changes from real human skin before and after applying a hydration lotion. (**D**) Skin impedance extracted from (C), at 100 kHz before and after applying the lotion. Reproduced from [70] with permission. © 2017 WILEY-VCH Verlag GmbH & Co. KGaA, Weinheim.

**Figure 24 sensors-19-04376-f024:**
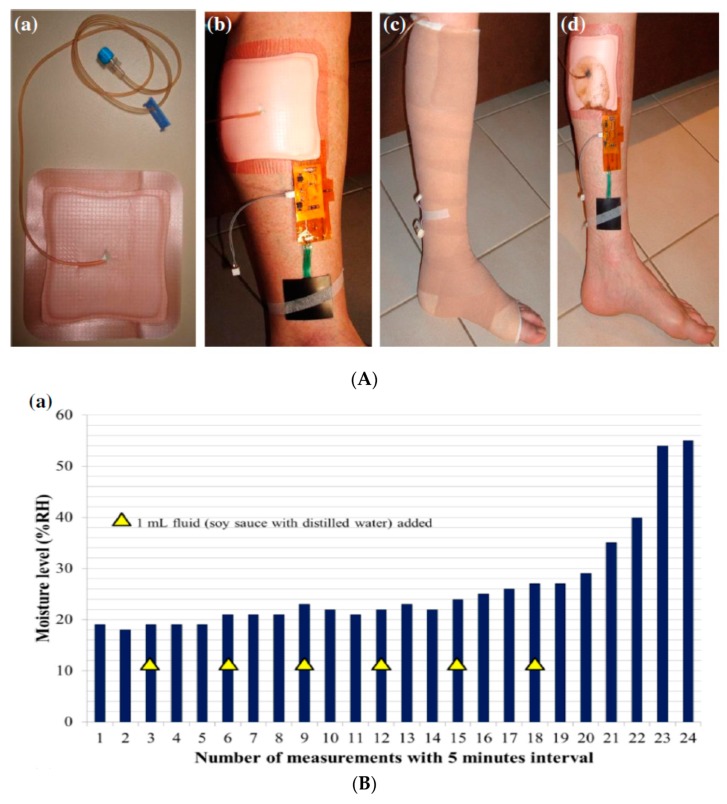
(**A**) (a) commercial Allevyn™ dressing mounted with tube, for injection of simulated exudates (obviously not intended for use on real patients). (b) Dressing and sensors in place on leg, before bandage application. (c) Bandage covering the whole device. (d) Dressing and sensors immediately after removal of bandages (the simulate exudate stain is clearly visible). (**B**) Measurement of moisture using the compression bandage shown in (a). Reproduced from [75]. Creative Commons Attribution License 4.0.

**Figure 25 sensors-19-04376-f025:**
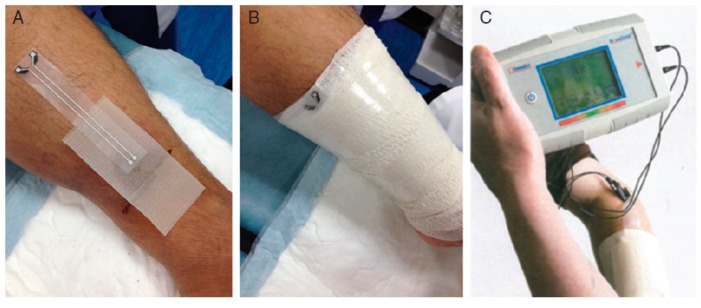
(**A**) WoundSense electrodes in direct contact with the wound, before dressing and (**B**) after dressing. (**C**) The WoundSense™ commercial meter. Reproduced from [77] with permission. © 2015 the authors. International Wound Journal published by Medicalhelplines.com Inc and John Wiley & Sons Ltd. Creative Commons Attribution-NonCommercial-NoDerivs Licence.

**Figure 26 sensors-19-04376-f026:**
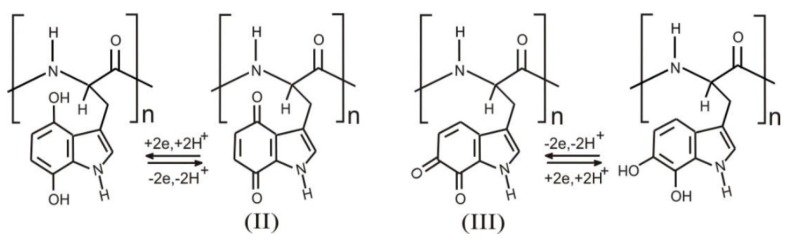
Oxidation products of a peptide homopolymer of tryptophan. After a first electrooxidation step, the fibers become electroactive due to the presence of the para- and orthoquinone groups. Reproduced from [79]. Creative Commons Attribution License.

**Figure 27 sensors-19-04376-f027:**
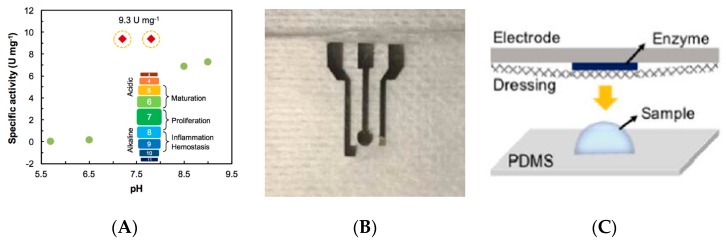
(**A**) Plots show the effects of pH on the specific activity of UOx. The red diamond markers denote higher activity of urate oxidase (UOx). The inset image represents the relationship between the wound healing stages and pH. (**B**) Screen-printed graphite + UA electrode on a dressing. (**C**) Experimental setup. Reproduced from [81]. © the author(s) 2018. Published by ECS. http://creativecommons.org/licenses/by/4.0/.

**Figure 28 sensors-19-04376-f028:**
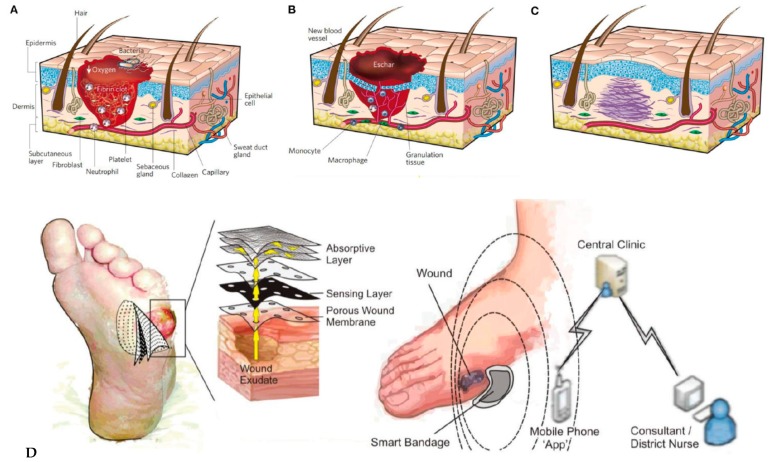
The three wound healing stages: (**A**) inflammation, (**B**) proliferation, and (**C**) remodeling. Reproduced from [89] with permission. Copyright © 2008, Springer Nature. (**D**) The general implementation and operation of a smart dressing. Reproduced from [79] Creative Commons Attribution License.

**Figure 29 sensors-19-04376-f029:**
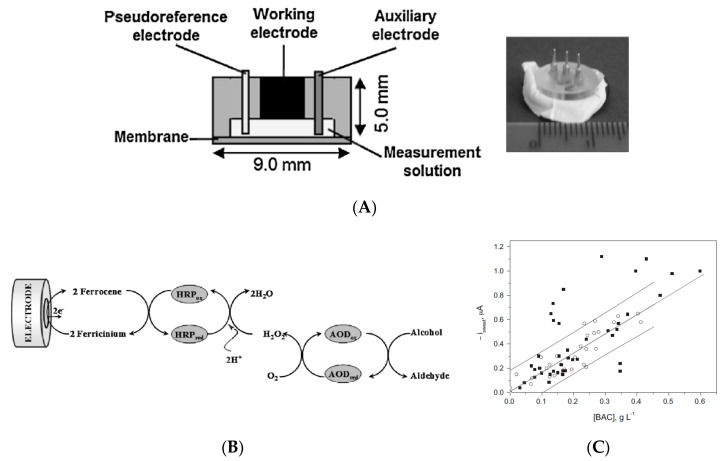
(**A**) Scheme of the various parts of the device (right: picture with contact electrodes). (**B**) Overall reactions involved in the sensing mechanism, which involves alcohol oxidase (AOD), a second enzyme (horse radish peroxidase—HRP) and ferrocene as mediator. (**C**) Correlation between the device output signal and blood alcohol concentration (BAC) values: (■) values measured with the biodevice in the single measurement mode, (○) values measured with the biodevice in the single measurement mode at 5 min (in both cases: n = 40 subjects). The straight lines show the corresponding BAC intervals obtained by the gas chromatography method. Reproduced from [91] with permission. Copyright © 2013 Elsevier B.V. All rights reserved.

**Figure 30 sensors-19-04376-f030:**
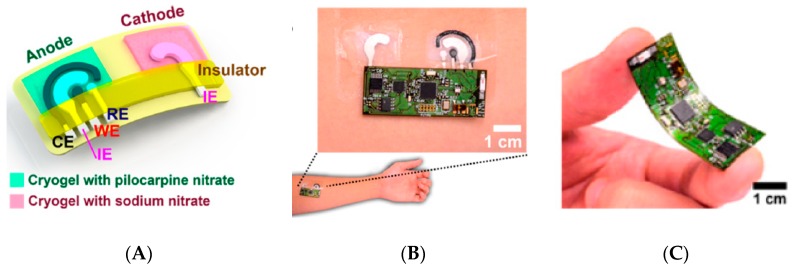
(**A**) Scheme of the iontophoretic flexible device, with the iontophoretic electrodes and the sensing electrodes. (**B**) Picture of the whole system (sensor with electronics) applied on the forearm of a subject. (**C**) Picture of the flexible wireless electronics. (**D**) Scheme of the constituents in the iontophoretic system (left) and in the amperometric electrode (right). (**E**) Experiments performed before (plot ‘a’) and after (plot ‘b’) consumption of 350 mL of beer measured on two different human subjects. Reproduced from [92] with permission. Copyright © 2016, American Chemical Society.

**Figure 31 sensors-19-04376-f031:**
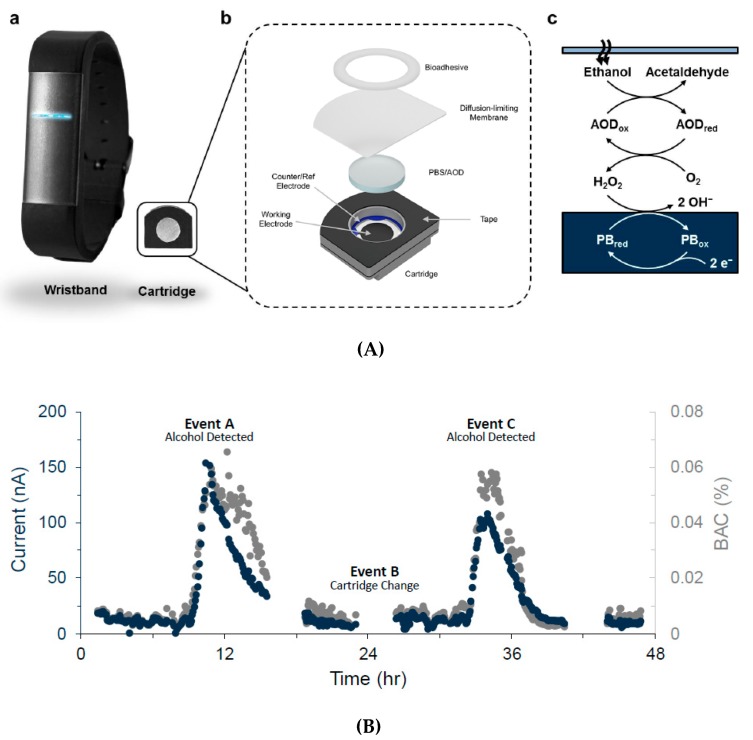
(**A**) The wristband sensor with its disposable cartridge (a); Scheme of the disposable sensor’s components (b); Amperometric detection principle, using alcohol oxidase and Prussian blue. (**B**) Current measurements using the developed device (grey) and derived equivalent blood alcohol concentration (black), for two periods of 24 h, interrupted by the necessary change of the disposable cartridge. Reproduced from [93]. Creative Commons Attribution License.

**Figure 32 sensors-19-04376-f032:**
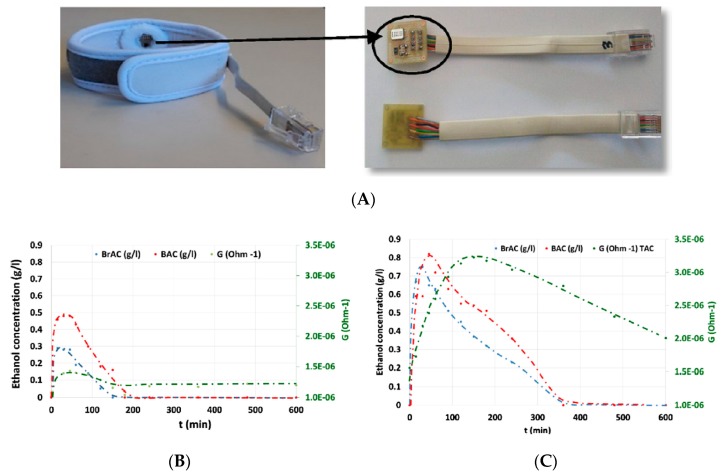
(**A**) Wristband carrying the SnO_2_ sensor and the electronics associated to it. (**B**) Comparison of blood (BAC), breath (BrAC) and transdermal (TAC) alcohol concentration for a maximum BAC of 0.5 g L^−1^. (**C**) Comparison of BAC, BrAC, and TAC for a maximum BAC of 0.5 g L^−1^. Reproduced from [94] with permission. © 2018 Elsevier B.V. All rights reserved.

**Figure 33 sensors-19-04376-f033:**
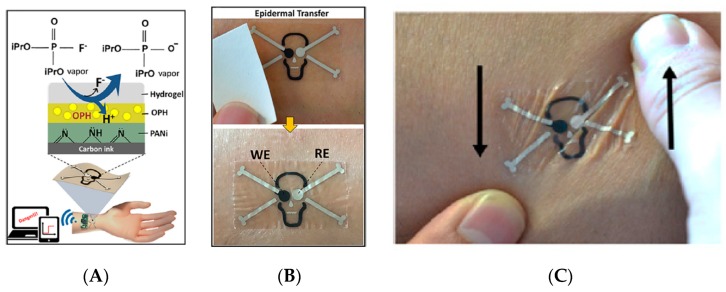
(**A**) Schematics of the potentiometric tattoo sensor working mechanism showing organophosphate hydrolysis on the OPH-modified electrode. Protons are released and protonate the polyaniline (PANi) layer. The data are transmitted wirelessly. The Ag/AgCl reference electrode is protected by a polyvinylbenzene (PVB) membrane containing NaCl. (**B**) Sensor transfer to the skin. (**C**) Resistance of the tattoo to mechanical strains. Reproduced from [96] with permission. © 2018 Elsevier B.V. All rights reserved.

**Figure 34 sensors-19-04376-f034:**
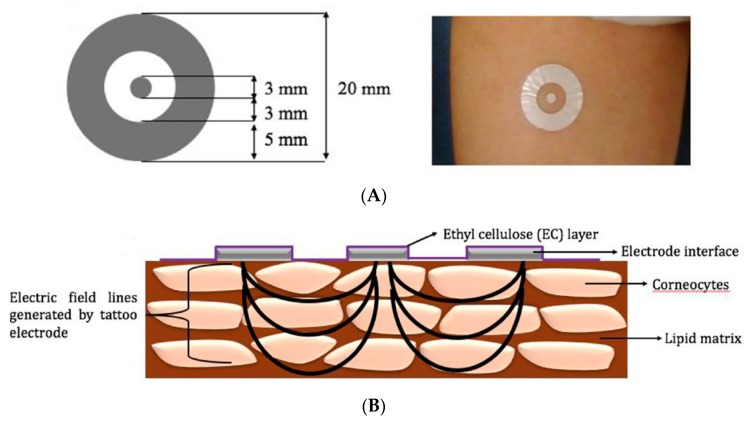
(**A**) Left: scheme and dimensions of the tattoo electrodes; Right: tattoo electrodes as applied on the inner forearm. (**B**) Side-profile scheme of the electrical field from the tattoo electrodes across the stratum corneum (SC). Reproduced from [98] with permission. © 2017 Wiley-VCH Verlag GmbH & Co. KGaA, Weinheim.

**Figure 35 sensors-19-04376-f035:**
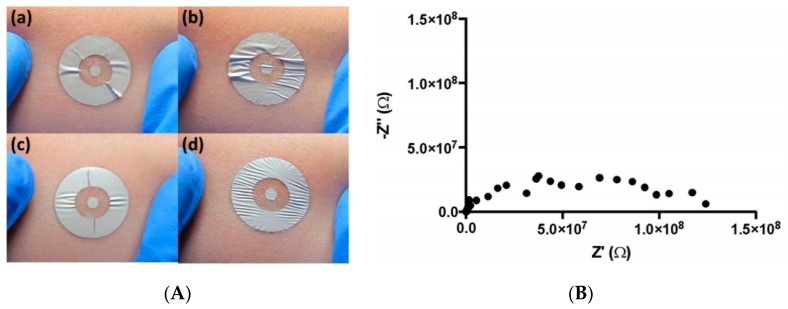
(**A**) Images of the thin flexible device upon stretching on skin, for (a) silver electrodes only, (b) silver + elastomer, (c) silver electrodes + porous acrylate adhesive and (d) silver + elastomer electrodes + porous acrylate adhesive. (**B**) Impedance (Nyquist plot) of the inner forearm, measured by the silver-elastomer tattoo device. Reproduced from [99] with permission from The Royal Society of Chemistry.

**Figure 36 sensors-19-04376-f036:**
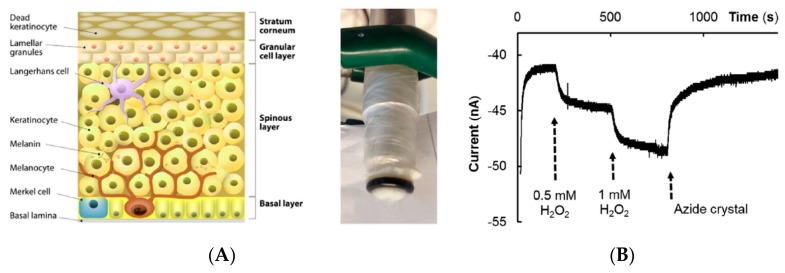
(**A**) (left) Scheme of the whole epidermis; (right) oxygen electrode covered with viable pig skin, used in this study. (**B**) Current delivered by the skin-covered oxygen electrode when immersed in phosphate buffer saline + sequential H_2_O_2_ addition. The catalase enzyme contained in the skin transforms H_2_O_2_ into O_2_, which diffuses back to the electrode. Azide, as catalase inhibitor, is added to attest that the current is due to the enzyme activity. Reproduced from [100] with permission. © 2017 Elsevier B.V. All rights reserved.

**Figure 37 sensors-19-04376-f037:**
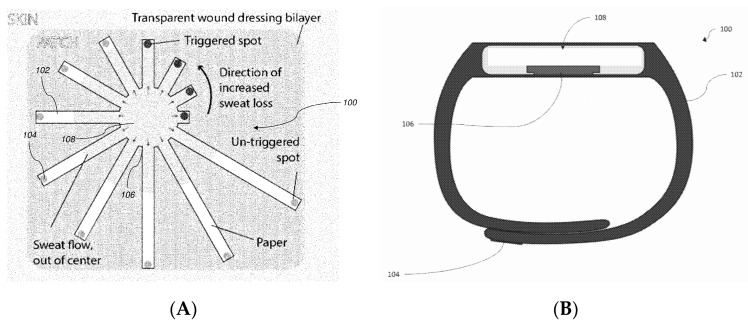
(**A**) The patch (100) claimed by [105] is based on cellulose fibers comprising radial channels (102) for sweat collection, carrying dyes at the end of the channels (104) for reading, and made of a wicking material (106) such as cellulose acetate or nitrocellulose laminated between two polymer films. The central collecting region (108) is made of the same material. (**B**) Wristband device claimed by Lansdorp et al. [106]. (100): whole device; (102) wristband; (104): wristband fastener; (106) sensor cartridge; (108) device body containing electronics.

**Figure 38 sensors-19-04376-f038:**
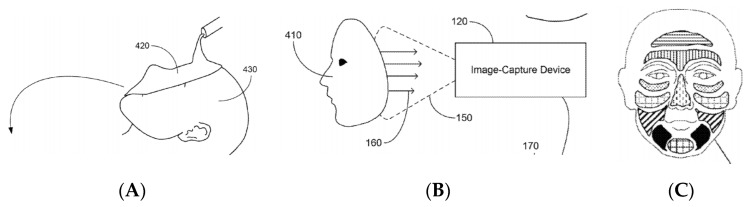
(**A**) Application of the settable material on the face of the subject. This settable material includes bacteria-sensitive regions. (**B**) After removing, the inner part of the mask (carrying the ELISA sensing layer) is analyzed with a camera which maps (**C**) the various regions of the subject’s face and quantify the bacteriome on each area. Ref. [107].

**Table 1 sensors-19-04376-t001:** Glossary of acronyms used in this review.

Acronyms	Definitions	Acronyms	Definitions
AgNW	Silver nanowire	OPH	Organophosphate hydrolase
AOD	Alcohol oxidase	PANi	Polyaniline
BAC	Blood alcohol concentration	PDMS	Polydimethylsiloxane
BrAC	Breath alcohol concentration	PEDOT:PSS	Poly(ethylene-3,4-dioxythiophene) blended with polystyrenesulfonate
CGM	Continuous glucose monitoring	PCA	Principal component analysis
CNT	Carbon nanotube	PCB	Printed circuit board
DFP	Diisopropyl fluorophosphate	PI	Polyimide
ECG	Electrocardiogram	PMMA	Poly(methylmethacrylate)
EIS	Electrochemical impedance spectroscopy	PSA	Pressure sensitive adhesive
ELISA	Enzyme linked immunosorbent assay	PU	Polyurethane
GC-MS	Gas chromatography coupled to mass spectroscopy	PVB	Polyvinylbenzene
GOx	Glucose oxidase	PVC	Polyvinylchloride
HRP	Horseradish peroxidase	RFID	Radiofrequency identification
IL-6	Interleukin-6	SAP	SAP Superabsorbent polymer
ISE	Ion selective membrane	SC	Stratum corneum
ISF	Interstitial fluid	SEBS	Styrene-ethylene-butylene-styrene
IS-FDSOI	Ion-sensitive fully depleted silicon on insulator	SEM	Scanning electron microscopy
ISFET	Ion sensitive field-effect transistor	SVM	Support vector machines
LA	Lactic acid	TAC	Transdermal alcohol concentration
LOD	Limit of detection	TDC	Tissue dielectric constant
LOx	Lactate oxidase	UA	Uric acid
MIP	Molecularly imprinted polymer	UOx	Uricase
NFC	Near field communication		
OECT	Organic electrochemical transistor

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
