# Peer review of "Recent Advances in Skin Chemical Sensors"

_sensors, 2019, doi:10.3390/s19204376_

Round 1
Reviewer 1 Report
The present work deals with a recent theme and of great prominence in the current literature. However, there are several excellent reviews published in the last 2-3 years with very similar scope, some of which were not even mentioned, such as doi: 10.1021/acsnano.7b04898, doi: 10.1016/j.trac.2019.115622, doi: 10.1002/adfm.201605271, doi: 10.3390/s19020363, doi:10.1002/adma.201705024, etc. The present work is quite comprehensive; however, for this kind of contribution, I expect a high content, judgment, suggestions, ideas from the analytical chemistry point of view. There are very few passages in the entire review about analytical chemistry performance, calibration/free calibration, implementation, validations, etc.
Additionally, some paragraphs/sections are too wordy, and it contains way too long sentences, which makes it challenging to follow the lecture.
I suggest that the authors better highlight other reviews by directing the readers to get more information on the topic and make clear the main contributions of this material in relation to the previously published works, as well as adding a higher critical analysis of the reviewed works (sensor performance: selectivity, stability, mechanical resistance, ....)
Author Response
We appreciated the constructive reviewer's comments.
In this revised version, we rewrote some parts so to be less wordy and more precise, and gave better descriptions of the cited works. Each time critical comments were not given in the previous version, we added some.
We also precised the added value of this review compared to previously published ones.
All changes on the revised version are underlined in yellow.
We hope that this revised version is now suitable for publication.
Please see the attachment.

Reviewer 2 Report
The paper is timely and comprehensive, covering the key areas of wearable sensors and some more niche areas (like drug testing in sweat). The authors are disciplined and have a succinct style throughout, focusing on the keyword wearables, thus correctly avoiding excessive reference to sensor fields that deviate from this central theme.
In general for wearable and on-body sensors, especially skin/sweat type sensors, the reviewer finds that there is often a lack of detail regarding the robustness and lifetime/reuseability of the proposed sensors i.e. on-body data sets are sometimes once off with little reproducibility data and it is not clear in many cases if the sensor can be reused or it is disposable.
Another weakness in articles in the literature is that the focus is on flexibility and comfort (which is ok) but the technical and more simple realities of sweat collection are often overlooked: How is the sweat collected? Is it fresh? Where does the spent sweat go? Is there mixing with old sweat/hysteresis? What is the total (sweat) capacity of the sensor if there is a reservoir?
Also, parallel validation studies (accuracy and precision, drift etc) should be insisted on. The author correctly highlights the drift issue with electrochemical sensors in particular. The underlying sensor transduction mode (enzymes etc) is often well know so the true novelty is the wearable nature of the proposed devises and the associated robustness/practical worth of the device (again as stated above).
Action: The author is asked to check all papers cited in the review to see again what the lifetime (reuseability/reversibility/conituous use/lifetime...) is of the sensors described and total sweat capacity and make a statement if this robustness-related data is/is not available. This has been done in some cases already but unclear in others cited and referred to in the current review.
Introduction: The author is correct to point out the fact that sweat analyte conc. must mirror or be related to blood levels and is correct to highlight papers where effort has been made in this respect - all new territory as extensive REAL time sweat data is only starting to come through, allowing comparison with blood levels (line 43). T
Section 3.1.1 warns about the difficulties of multicomponent analysis and states there isnt a one size fits all. A timely point.
The author does well to point these out in the review - where the information exists, which is helpful.
The commercial slant of the research under investigation is referred to which is timely: patent section and reference to glucose as the biggest current market.
Action: Some of the plots of data shown (eg Fig 6, 16 etc - please check ALL plot Figures) have no x axis label so it is not clear what the shown data represents.
Typo on line 343 'deveopping'.
On p. 13 a timely statement that LOD is not critical in many on-body sensors is timely, rather operational range is important. LOD has a far higher importance in environmental and industrial sensing.
Action: Line 440: Please provide evidence (references?) for the statement that Cl or Na is sweat is an indicator of dehydration - where has this been clinically proven?
The author is commended for referring to often overlooked (an unglamorous) aspects of such sensors namely reference electrodes (line 547) .
The conclusions and perspectives are insightful.
Author Response
We appreciated the constructive reviewer's comments.
In this revised version, we add more critical comments and more details for each cited work, so to point out not only the "comfort" characteristics but also the analytical ones when available or realized under conditions which allow comparison. We also precised robustess, reprodutibility, stability, or the way(s) sweat is collected (when applicable).
We rewrote the conclusion and add paragraphs when needed.
We also add the suggested particular comments made in the second part of your review.
All changes are highlighted in yellow.
We hope that our manuscript is now suitable for publication.
Please see the attachment.

Reviewer 3 Report
This review summarized the development of skin based bio-chemical sensors in the latest years. This topic is very important for both healthcare monitoring and disease diagnosing and a lot of progress have been made in recent years. The authors organized the work by specific applications, which includes sweat analysis, skin hydration, monitoring of skin wounds, alcohol or drugs detection, and general skin status, reactive oxygen species and skin microbiota. In each section, related methods and applications have been illustrated. Overall, this review would be a great contribution to Sensors since it covers most of works that were developed in last 3 to 5 years. Some remarks are suggested to be considered as following:
Abstract needs to be revised. It should be more clear and straightforward to summarize the importance of this work and the applications that will be covered in this review. I did not consider it as a good idea to starting the introduction with a very specific example of continuous glucose monitoring (CGM). It is an important example of skin chemical sensor, but not suitable to be placed in the first paragraph of the introduction. I would recommend to provide high-level introduction to talk about the importance of skin chemical sensors. In the first paragraph of section 3 “Discussion”, the authors discussed the previous reviews on the topic of wearable skin chemical sensors and what will be covered in this work. It would be better to move this paragraph to introduction, since it gives readers better understanding of the structure as well as the content of this review. Moreover, since the authors have already mentioned the previous reviews done by other researchers on wearable skin chemical sensors, it would be good to add several sentences to clarify what the differences of this work from the previous reviews. One typo. On page 30, the section number of “Patents” should be 3.6 instead of 3.7. Some sentences are way too long to understand. Short sentences are usually preferred.
Author Response
We appreciate the constructive comments of the reviewer.
We made significant changes which are highlighted in yellow. The abstract and the introduction were rewritten. Comments concerning some works were added to be more critical.
We believe that we answered to the reviewer's comment, and hope that our manuscript is now suitable for publication.
Please see the attachment.

Round 2
Reviewer 1 Report
Accept.